# Phenotypic Characterization of Arabidopsis Ascorbate and Glutathione Deficient Mutants under Abiotic Stresses

Minh Thi Thanh Hoang [1,2,*,†] , Mai Thi Anh Doan [1,2,†], Thuong Nguyen [1,2,†], Dong-Phuong Tra [1,2], Thanh Nguyen Chu [1,2], Thi Phuong Thao Dang [2] and Phuong Ngo Diem Quach [1,2]

[1] Faculty of Biology and Biotechnology, University of Science, Ho Chi Minh City 700000, Vietnam; dtanhmai@gmail.com (M.T.A.D.); nguyenthuongnt1101@gmail.com (T.N.); tdphuong@hcmus.edu.vn (D.-P.T.); cnthanh14@gmail.com (T.N.C.); qndphuong@hcmus.edu.vn (P.N.D.Q.)

[2] Laboratory of Molecular Biotechnology, University of Science, Vietnam National University, Ho Chi Minh City 700000, Vietnam; dtpthao@hcmus.edu.vn

[*] Correspondence: httminh@hcmus.edu.vn

[†] Minh Thi Thanh Hoang, Mai Thi Anh Doan and Thuong Nguyen contributed equally to this article.

**Abstract:** Ascorbic acid (AsA) and glutathione (GSH) are considered important factors to protect plants against abiotic stress. To investigate whether altered endogenous GSH and AsA affect seed germination, plant performance and the abiotic stress tolerance, GSH deficient mutant *cad2-1* and AsA-deficient mutants (*vtc2-4* and *vtc5-2*) were phenotypically characterized for their seed germination, shoot growth, photosynthetic activity and root architecture under abiotic stresses. The *cad2-1*, *vtc2-4* and *vtc5-2* mutants showed a decrease in osmotic and salt stress tolerance, in sensitivity to ABA during seed germination, and in plant performance under severe abiotic stresses. GSH deficiency in the *cad2-1* plants affected plant growth and root development in plants exposed to strong drought, oxidative and heavy metal stress conditions. Plants with lower GSH did not show an increased sensitivity to strong salt stress (100 mM NaCl). In contrast, the mutants with lower AsA enhanced salt stress tolerance in the long-term exposures to strong salt stress since they showed larger leaf areas, longer primary roots and more lateral root numbers. Limitations on AsA or GSH synthesis had no effect on photosynthesis in plants exposed to long-term strong salt or drought stresses, whereas they effected on photosynthesis of mutants exposed to $CdCl_2$. Taken together, the current study suggests that AsA and GSH are important for seed germination, root architecture, shoot growth and plant performance in response to different abiotic stresses, and their functions are dependent on the stress-inducing agents and the stress levels.

**Keywords:** abiotic stress tolerance; ascorbate (AsA); *cad2-1*; glutathione (GSH); leaf area; photosynthesis; root architecture; seed germination; *vtc2-4*; *vtc5-2*

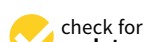

## 1. Introduction

Abiotic stresses, such as salinity, drought, temperature extremes and heavy metals toxicity, are major factors in limiting plant growth and decreasing crop productivity. The exposure of plants to unfavorable environmental conditions increases the production of reactive oxygen species (ROS) such as, superoxide ($O^{2\bullet-}$), hydrogen peroxide ($H_2O_2$), and hydroxyl radical ($OH^\bullet$) [1]. The production of excessive ROS in plant cells leads to oxidative cellular damage, which ultimately affects the plant growth and productivity [2]. To protect themselves from adverse conditions, plants have evolved a number of cellular defense mechanisms, including the employment of antioxidants such as ascorbate (AsA), glutathione (GSH) and tocopherols, as well as ROS-detoxifying enzymes such as superoxide dismutases, peroxidases and catalases [3]. Among these stress-related molecules, the two soluble antioxidants, AsA and GSH, are central components of the AsA-GSH cycle, which regulates the cellular redox homeostasis, and are involved in plant tolerance against abiotic and biotic stresses [4].

AsA plays important roles in stress tolerance, cellular redox regulation and redox signaling [4,5]. An increase in AsA content contributes to abiotic stress tolerance in Arabidopsis and tobacco [6,7]. Previous studies have shown that overexpressing genes of the AsA biosynthetic pathway lead to an increase in AsA content, and enhances abiotic stress tolerance [8–10]. Plants with lower AsA content displayed reduced tolerance to salt, heat and high light stress [11–13]. Furthermore, AsA also involves in many biological processes, including cell wall biosynthesis, elongation, cell division, iron uptake, hormone biosynthesis, anthocyanin accumulation, and the xanthophyll cycle [12,14–17]. AsA is synthesized on the plant mitochondrial inner membrane, and then distributed throughout different cellular compartments/organelles, such as apoplast, vacuoles, mitochondria, cytosol and chloroplasts. In higher plants, L-galactose pathway, a major ascorbic acid biosynthesis pathway, was characterized and confirmed by genetic analysis in Arabidopsis [18]. The central reaction in this pathway is the conversion of GDP-L-galactose to L-Galactose 1-P by the enzyme GDP-L-galactose phosphorylase encoded by the *Vitamin C-defective 2* (*VTC2*) and *VTC5* genes [19]. Two At4g26850 (*VTC2*) T-DNA insertion mutants (SALK_146824 or SAIL_ 769_H05 named *vtc2-4*) showed severe reduction of AsA biosynthesis, producing only 20–30% of the wild-type (WT) ascorbate level while two At5g55120 (*VTC5*) T-DNA insertion mutants (*vtc5-1* and *vtc5-2*) displayed a slight reduction of AsA level with 80% of the WT ascorbate level [5,15,19]. The *vtc2-4* (SAIL_ 769_H05) and *vtc5-1*(SALK_000989) mutants showed no phenotypic growth difference to WT on standard conditions [5,19]. Although the physiological function of AsA in abiotic stress tolerance has been discovered, the mechanism of AsA involvement in the control of plant growth and development under abiotic stress conditions is largely unknown. Therefore, the present study was designed to explore further the effects of altered endogenous AA on germination, shoot growth, root development and photosynthesis in response to abiotic stresses by phenotyping of Arabidopsis AsA-deficient mutants with moderate (*vtc5-2*) or very low (*vtc2-4*) AsA content, compared to WT.

GSH is a low molecular weight thiol tripeptide (γ-glutamyl-cysteinyl-glycine) with multiple functions in plants. It is involved in cell differentiation, cell growth/division, cell death, detoxification and expression of stress responsive genes [1]. GSH is an important component of the plant antioxidant system, which is involved in abiotic stress signaling and tolerance [20]. An increased GSH level is commonly observed in plants under stress, and exogenously applied GSH can improve abiotic stress tolerances in plants [21]. The metabolism of glutathione has been extensively characterized in plants and other organisms. Glutathione is synthesized in a two-step process catalyzed by the gammaglutamyl cysteine synthase (GSH1) and the glutathione synthase (GSH2) [22]. GSH acts as a crucial regulator of normal plant metabolism since the loss-of-function of GSH biosynthesis genes causes an embryonic lethal phenotype [23]. The cadmium-sensitive 2 mutant (*cad2-1*) carries a 6 bp deletion within an exon of GSH1, has about 20–30% of the WT GSH content and exhibits the hypersensitivity to cadmium [24]. However, the *cad2-1* displayed no difference to the WT throughout vegetative development [25]. Another mutation at AtGSH1 is *pad2-1* mutant, which contains only 22% of the WT level of glutathione, showing enhanced sensitivity to pathogens [26]. The *cad2-1* and *pad2-1* showed the negative impacts on leaf area when it was exposed long-term on high salt and sorbitol treatment [22]. In contrast, the *pad2-1* mutant plants had a lower survival rate compared to WT plants after a two-week drought treatment [27].

Although many studies on mutants and /or transgenic plants with altered levels of AsA or GSH levels proved the roles of AsA and GSH in abiotic stress responses, the functions of AsA and GSH in the control of germination, plant growth, photosynthesis and root development under abiotic stress conditions require further investigations. Furthermore, there are several contradictory results on the phenotypes of glutathione deficient mutants in response to abiotic stress conditions [20]. The apparent contradictory findings among these studies may have resulted from variations in stress conditions and scoring system [20]. Moreover, several previous studies that used a *vtc2-1* mutant line containing

an independently cryptic mutation, which affected the growth and physiological responses in different conditions, should be re-evaluated [5]. In addition, although AsA and GSH are both main important antioxidants for AsA-GSH cycle, the GSH- and AsA-related biosynthetic pathways in the response to abiotic stress conditions used to be investigated separately. Therefore, to further elucidate the physiological effects of altered endogenous AsA and GSH levels in the response of Arabidopsis to abiotic stress conditions, this study was designed for: (1) characterization of the seed germination and leaf area phenotypes in different AsA-deficient mutants (*vtc2-4* and *vtc5-2*), and the GSH-deficient mutant *cad2-1* under abiotic stresses; (2) measurement of the photosynthetic activity of AsA and GSH-deficient mutants; and (3) determination of the roles of AsA and GSH biosynthesis in the root development under control and abiotic stresses (salt, drought, oxidative stresses and $CdCl_2$ toxicity). The findings demonstrate that AsA and GSH play various roles in plant abiotic stress tolerance, but their functions are dependent on stress-inducing agents and stress levels.

## 2. Results

### 2.1. AtGSH1 and AtGSH2 Can Complement Yeast Δgsh1 and Δgsh2 Mutants

In the glutathione-biosynthetic-deficient mutants, the yeast mutants lacking the *GSH1* or *GSH2* genes involved in two-step glutathione synthesis (Δ*gsh1* and Δ*gsh2*) were reported to be sensitive to oxidative stress and $CdCl_2$ [28,29]. To understand the function of GSH1 and GSH2 in response to abiotic stress conditions, yeast Δ*gsh1* and Δ*gsh2* strains were grown on different abiotic stress conditions was observed. The current study showed that yeast GSH-deficient mutants also displayed sensitivity to hyper-osmotic stresses (Figure 1). Therefore, this experiment was performed to determine whether the heterologous expression of AtGSH1 on Δ*gsh1* and the expression of AtGSH2 on Δ*gsh2* can rescue the growth defect phenotype of glutathione-deficient mutants on high concentration of $CdCl_2$, NaCl or sorbitol. Both AtGSH1-expressing Δ*gsh1* yeast cells and AtGSH2-expressing Δ*gsh2* yeast cells were grown to the same level as WT cells on medium containing 100 μM $CdCl_2$ or hypertonic medium containing 1 M NaCl or 1.5 M Sorbitol (Figure 1). These results indicated that the expression of AtGSH1 on Δ*gsh1* and AtGSH2 on Δ*gsh2* restored the cadmium and hyper-osmotic tolerance of GSH-deficient mutants.

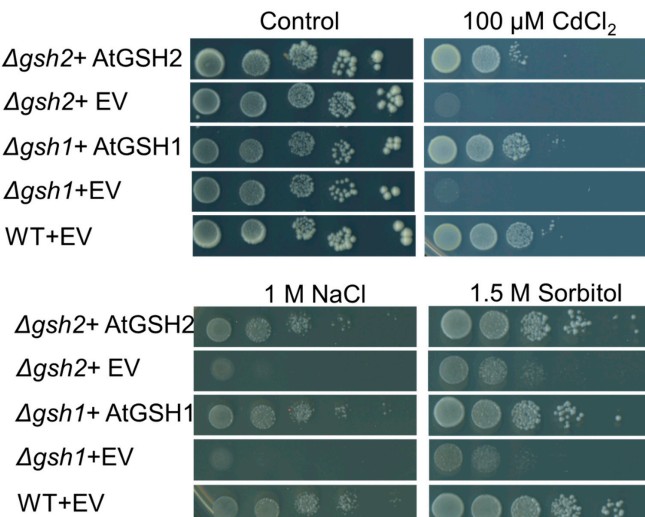

**Figure 1.** Complementation of Δ*gsh1* and Δ*gsh2* by AtGSH1 and AtGSH2, respectively under heavy metal and hypertonic conditions. An empty vector pDR195 (EV), a vector expressing AtGSH1 (AtGSH1), a vector expressing AtGSH2 (AtGSH2) were transformed into yeast WT cells or yeast mutant cells Δ*gsh1* or Δ*gsh2*. 10-fold serial dilutions of the transformed yeast cells were dropped onto different media. Photographs were taken after 3–4 days incubated at 30 °C. Similar results were observed in three independent experiments.

## 2.2. The Lower GSH and AsA Concentration in Arabidopsis Showed a Reduced Salt, Osmotic Stress Tolerance and an Increased Insensitivity to ABA during Germination

AtGSH1 and AtGSH2 enhanced salt and osmotic stress tolerance in yeast. This observation raised a question about the functions of AtGSH1 in response to abiotic stress conditions in different growth stages of plants such as germination and vegetative growth. To determine the roles of GSH during germination in response to different environmental stresses, germination tests under salt stress and osmotic stress were performed. First, seed germination in response to salt stress was investigated in *cad2-1* and WT by sowing their seeds on $\frac{1}{2}$ MS medium supplemented with different concentrations of NaCl (0, 50, 100, 150, 170 and 200 mM). As shown in Figure 2A, the WT and *cad2-1* mutant seed germination rates were similar under standard condition and up to 100 mM NaCl treatment. When being exposed to 150 mM NaCl and higher NaCl concentrations, the germination rates of *cad2-1* seeds were significantly lower than that of WT seeds on the second day after sown on plates (Figure 2A). These results suggest that *cad2-1* mutant was more sensitive to salt stress than WT. Second, the current study investigated the *cad2-1* mutant seed germination in response to osmotic stress. The WT seeds and the *cad2-1* seeds were sown on $\frac{1}{2}$ MS medium supplement with 0, 200, 300, 400, 500 and 600 mM sorbitol. The WT seeds showed a higher germination rate than the *cad2-1* mutant in the presence of sorbitol. In 300 mM sorbitol, the *cad2-1* seeds showed a 21–22% reduction in germination rate than WT seeds after 6 days (Figure 2B). These results indicated that the *cad2-1* mutant was more sensitive to osmotic stress than WT during seed germination stage. Taken together, the lower glutathione concentration in Arabidopsis increased the sensitivity to salt and osmotic stress at germination stage.

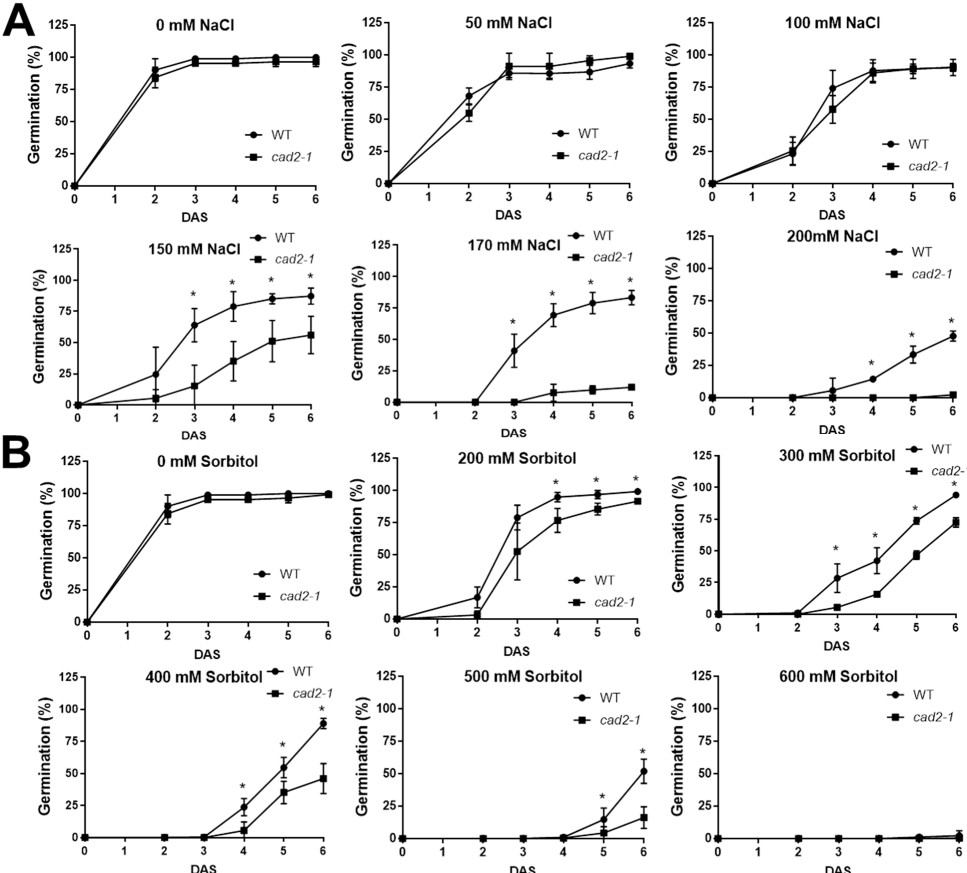

**Figure 2.** The *Arabidopsis thaliana* GSH-deficient mutant *cad2-1* is more sensitive to salt and osmotic stress conditions during germination. (**A**). Germination percentages of WT and *cad2-1* seeds under salt stress. (**B**). Germination percentages of WT and *cad2-1* seeds under osmotic stress. Germination percentages were counted at the indicated times. Data are shown as the means ± SD of three independent experiments using 50–100 seeds of each genotype. Experiments were repeated at least twice with similar results. Asterisks indicate significant differences (Student's *t*-test; * $p < 0.05$).

AsA is not only an important component of human nutrition but an antioxidant and $H_2O_2$-scavenger that defends plants against abiotic stress [9]. The maintenance of AsA level is required for oxidative stress tolerance in Arabidopsis [9]. Germination with various NaCl or sorbitol concentrations on WT seeds showed that 150 mM NaCl or 300 mM sorbitol decreased germination rates by about 65% or 72%, respectively after 72 h in above experiments. Therefore, to determine whether the lower AsA concentration in Arabidopsis alters NaCl or sorbitol sensitivity during germination, AsA-deficient mutants were germinated in the presence of 150 mM NaCl or 300 mM Sorbitol, and the germination rates were evaluated after 72 h. In control treatment (no stressor), the germination rates of *vtc2-4* and *vtc5-2* mutant seeds were similar to that of WT seeds. The germination rate of *vtc2-4* seeds was much lower than *vtc5-2* and WT seeds in the presence of NaCl or sorbitol. After 72 h of NaCl treatment, the germination rates of *vtc2-4* and *vtc5-2* mutants were 5.56% and 47.78%, respectively, while that of WT was 58.89% (Figure 3). Similarly, the germination rates of *vtc2-4* and *vtc5-2* mutants after 72 h under sorbitol treatment were 16.67% and 52.22%, respectively, while that of WT was 83.33% (Figure 3). Together, the current study showed that the deficiencies in AsA synthesis decreased the salt and osmotic tolerance during seed germination stage.

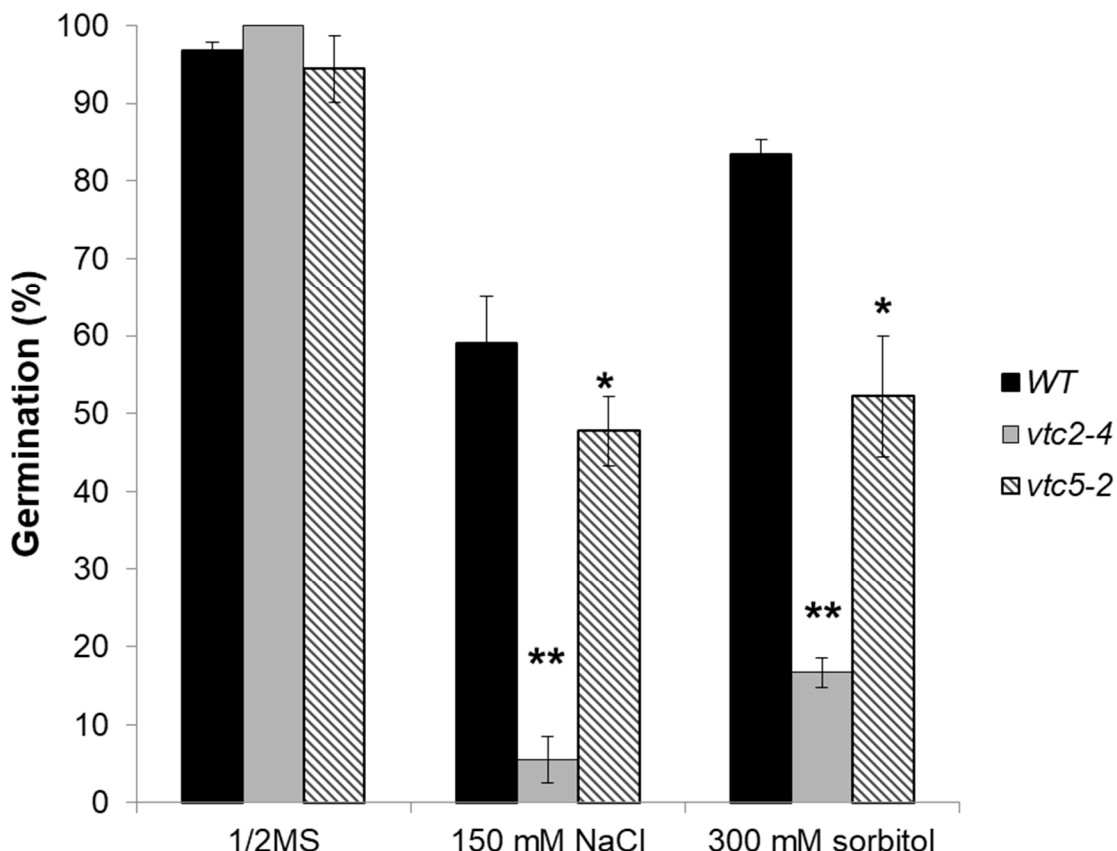

**Figure 3.** Seeds of *Arabidopsis thaliana* AsA-deficient mutants are more sensitive to salt and osmotic stress conditions during seed germination. Seeds of WT, *vtc2-4* and *vtc5-2* were assayed for germination on control (black), or in response to with 150 mM NaCl (gray) or 300 mM sorbitol (stripped). Germination percentages were recorded after 3 days sowing. Data are shown as the means ± SD of three independent experiments using 50 seeds of each genotype. Experiments were repeated at least twice with similar results. Asterisks indicate significant differences (Student's *t*-test; * $p < 0.05$ and ** $p < 0.01$).

Abscisic acid (ABA) plays important roles in abiotic stress response and regulation of seed germination [30]. Therefore, this study investigated whether the seed germination was affected by exogenous ABA in the GSH-deficient mutant *cad2-1* and AsA-deficient *vtc2-4* and *vtc5-2* mutants. The WT, *cad2-1*, *vtc2-4* and *vtc5-2* seeds were sown on $\frac{1}{2}$ MS

medium supplement with 0, 2.5, 5, 10 and 15 μM ABA. In contrast to the effect of sorbitol or salt, *cad2-1*, *vtc2-4* and *vtc5-2* were less sensitive to the inhibitory effect of ABA on seed germination compared to WT. All of the mutants had higher germination rate than WT, when the seeds were treated with different concentrations of ABA (Figure 4 and Supplementary Table S1). The current study indicates that GSH-deficient mutant and AsA-deficient mutants displayed the reduction of salt and osmotic tolerance during germination and an increased insensitivity to ABA, the key phytohormone in maintaining the seed dormancy stage.

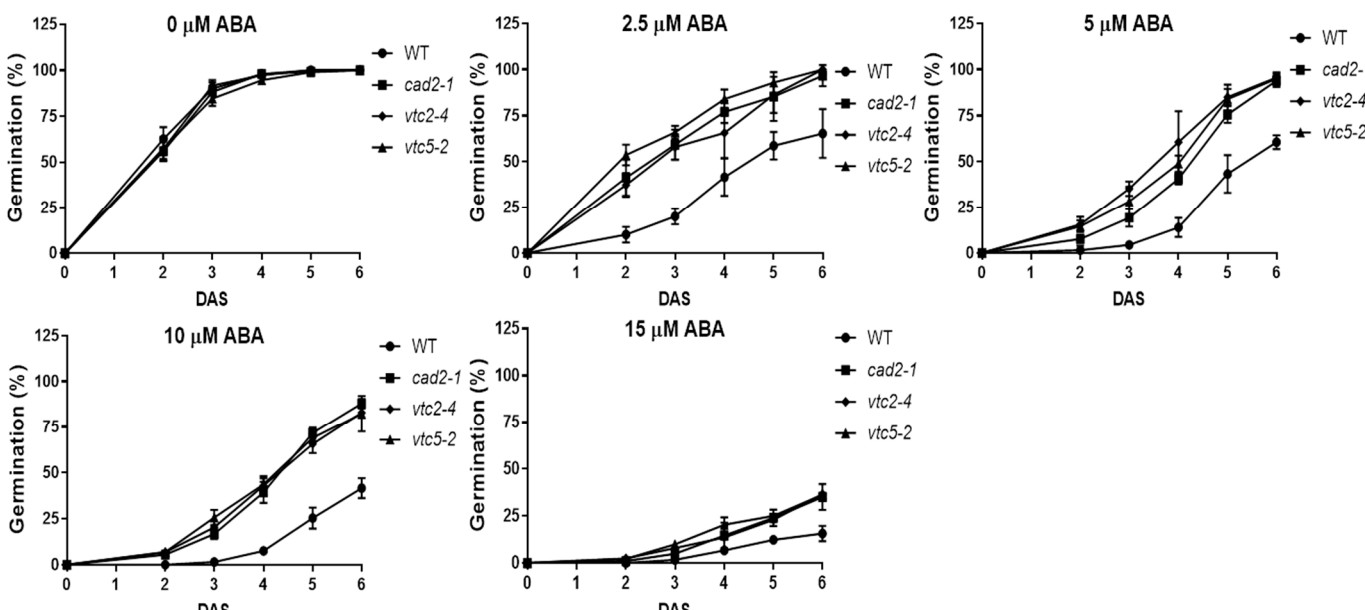

**Figure 4.** Seeds of *Arabidopsis thaliana cad2-1*, *vtc2-4* and *vtc5-2* are more resistant to ABA-mediated inhibition of seed germination. Germination rates (%) were recorded at the indicated time-points. Data are shown as the means ± SD of three independent experiments using 50–100 seeds of each genotype. The experiments were repeated at least twice with similar results.

### 2.3. Leaf Areas of AsA and GSH-Deficient Mutants Were Differently Affected by Long-Term Exposures to Various Abiotic Stresses

To investigate the importance of AsA and GSH in leaf growth under different abiotic conditions, the AsA-deficient *vtc2-4* and *vtc5-2* mutants and the GSH-deficient *cad2-1* mutant were exposed to various long-term abiotic stresses and leaf areas of plants were measured and analyzed. Shoot fresh weight was also analyzed but showed the same trends as leaf area (Supplementary Figure S1). Under control condition ($\frac{1}{2}$ MS), there was no significant difference among WT, *cad2-1*, *vtc2-4* and *vtc5-2* (Figure 5). After 18–21 days exposed to salt stress (100 mM NaCl), leaf areas of the *vtc2-4* and *vtc5-2* mutants were 91% and 82% larger than that of WT plants, respectively, while the results showed no difference between WT and *cad2-1* plants. Under osmotic stress (225 mM sorbitol), leaf area of the *cad2-1* was reduced by 16% compared to WT, while leaf area of *vtc2-4* and *vtc5-2* showed no significant difference. Leaf areas of *cad2-1*, *vtc2-4* and *vtc5-2* were reduced by 77%, 18% and 31%, respectively, compared to WT under oxidative stress condition (1 mM $H_2O_2$) (Figure 5). When exposed to $CdCl_2$, *cad2-1* leaf area decreased significantly, up to 81% compared to WT. GSH is involved in cell cycle regulation in leaves of cadmium-exposed plants [31]. The stronger leaf growth inhibition observed in *cad2-1* mutants compared to WT plants upon prolonged Cd exposure underlines the importance of GSH in plant defense against Cd. Interestingly, leaf areas of *vtc2-4* and *vtc5-2* were larger than WT when they were exposed to $CdCl_2$, 64% and 97%, respectively. Together with the results observed under oxidative stress, these results confirmed the critical functions of ASA and GSH: both

are important antioxidants in redox balance mechanism, while GSH alone is a key player in heavy metal ($CdCl_2$) detoxification. These data indicated that an impaired GSH synthesis adversely affected leaf growth in plants exposed long-term osmotic, oxidative and heavy metal stresses but not salt stress while AsA deficiency negatively affected leaf growth in plants exposed to salt and heavy metal stresses but not oxidative stress.

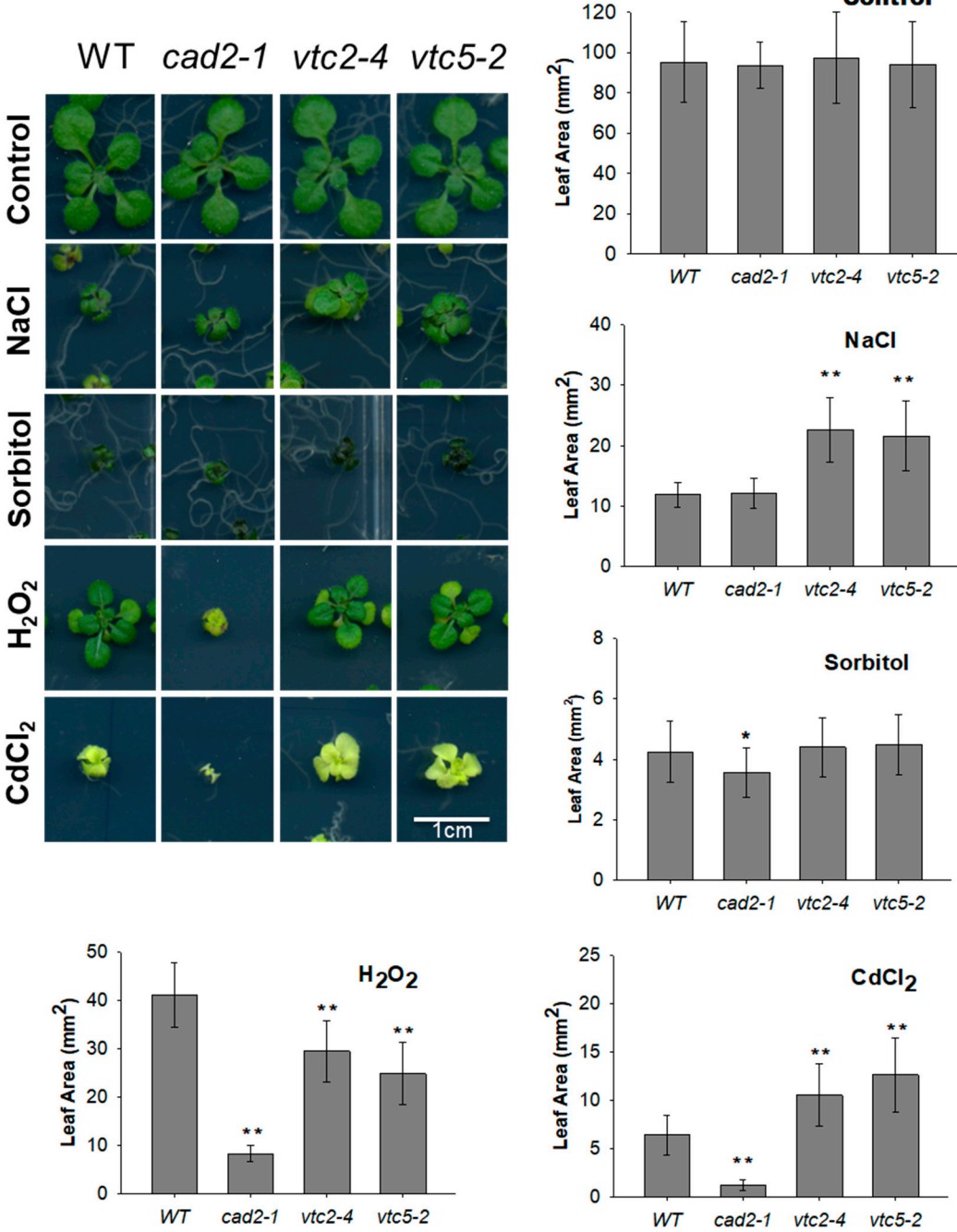

**Figure 5.** Rosette leaf area of the *cad2-1*, *vtc2-4*, *vtc5-2* mutants and WT under normal conditions or different strong stress conditions. The *cad2-1*, *vtc2-4* and *vtc5-2* mutant as well as the WT were exposed to different strong abiotic stress conditions: salt stress (100 mM NaCl), osmotic stress (225 mM sorbitol), oxidative stress (1 mM $H_2O_2$), and heavy metal stress (40 µM $CdCl_2$) for 18–21 days. Bar graphs represent mean of treatments, with error bars indicate Standard Deviation (N = 25–35). Experiments were performed twice with similar results. Asterisks indicated the statistical significance of differences between WT and mutants by Student's *t*-test: * $p < 0.05$ and ** $p < 0.01$. White bar in the photograph corresponds to 1 cm.

### 2.4. GSH-Deficient Mutants Had a Significantly Lower Photosynthesis than the WT under Oxidative and Heavy Metal Stresses

Photosynthesis is one of the major determinants of plant growth and yield formation [32]. To investigate the link between the reduction of AsA and GSH level and photosynthesis under abiotic stress, maximum PSII efficiency ($F_v/F_m$) of rosette leaves of the WT, *cad2-1*, *vtc2-4* and *vtc5-2* grown for 18-21 days in different abiotic conditions was measured (Figure 6). Arabidopsis WT and mutants plants grown on control conditions (no stressor) showed $F_v/F_m$ values around 0.77. Maximum PSII efficiencies of WT and mutants were not significantly different in control conditions versus salt stress and osmotic stress. However, in oxidative stress, *cad2-1* Fv/Fm was reduced by 26% compared to WT, while Fv/Fm of *vtc2-4* and *vtc5-2* was not dramatically affected, indicating the importance of GSH in oxidative stress tolerance. In heavy metal stress, the results showed that Fv/Fm of *cad2-1* and *vtc2-4* was respectively 58% and 67% compared to WT, while Fv/Fm of *vtc5-2* showed no difference. While Fv/Fm of *vtc2-4* in heavy metal stress was affected, the PSII efficiency of *vtc5-2* was similar to WT. In conclusion, the *cad2-1*, *vtc2-4* and *vtc5-2* mutants responded similarly in terms of photosynthetic activity in strong stress level of NaCl and sorbitol, whereas the photosynthetic activity of the *cad2-1* was affected under $CdCl_2$ and $H_2O_2$, and photosynthetic activity of the *vtc2-4* was significantly decreased under $CdCl_2$.

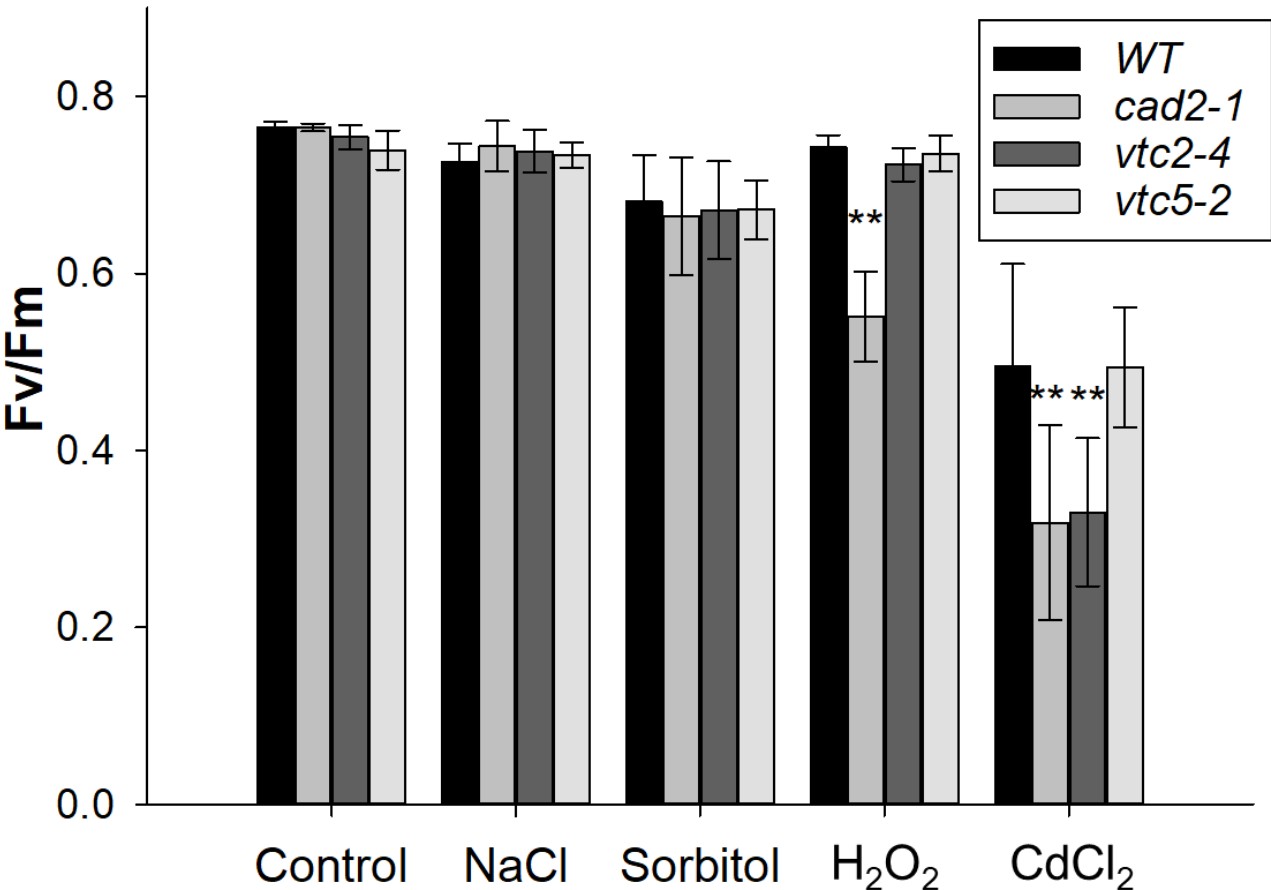

**Figure 6.** Photosynthetic activity of Arabidopsis WT, *cad2-1*, *vtc2-4* and *vtc5-2* mutants grown under control and different abiotic stress conditions. Maximum PSII efficiency ($F_v/F_m$) measured on rosette leaves of the WT, *cad2-1*, *vtc2-4* and *vtc5-2* grown on different strong abiotic stress conditions: salt stress (100 mM NaCl), osmotic stress (225 mM sorbitol), oxidative stress (1 mM $H_2O_2$), and heavy metal stress (40 μM $CdCl_2$) for 18–21 days. FluorCam FC 800-O (Photon Systems Instruments) was used to estimate the maximal photochemical efficiency of PSII [$F_v/F_m = (F_m - F_o)/F_m$] which revealed PSII activity. Data are shown as the means ± SD (*n* = 10–25). Experiments were performed twice with similar results. Asterisks indicate values significantly different from those of the WT (Student's *t*-test, ** *p* < 0.01).

### 2.5. GSH and AsA Deficiency Differentially Altered Root Architecture of Plants Exposed to Long-Term Abiotic Stresses

In order to determine the effects of abiotic stresses on Arabidopsis root architecture of GSH-deficient mutants and AsA-deficient mutants, Arabidopsis seeds were germinated and grown on control or strong stress media containing 100 mM NaCl or 225 mM sorbitol or 1.0 mM $H_2O_2$ or 40 µM $CdCl_2$. Primary root length and lateral root (LR) number were identified from the images of the plantlets at 13–14 day (Figures 7 and 8 and Supplemental Figure S2).

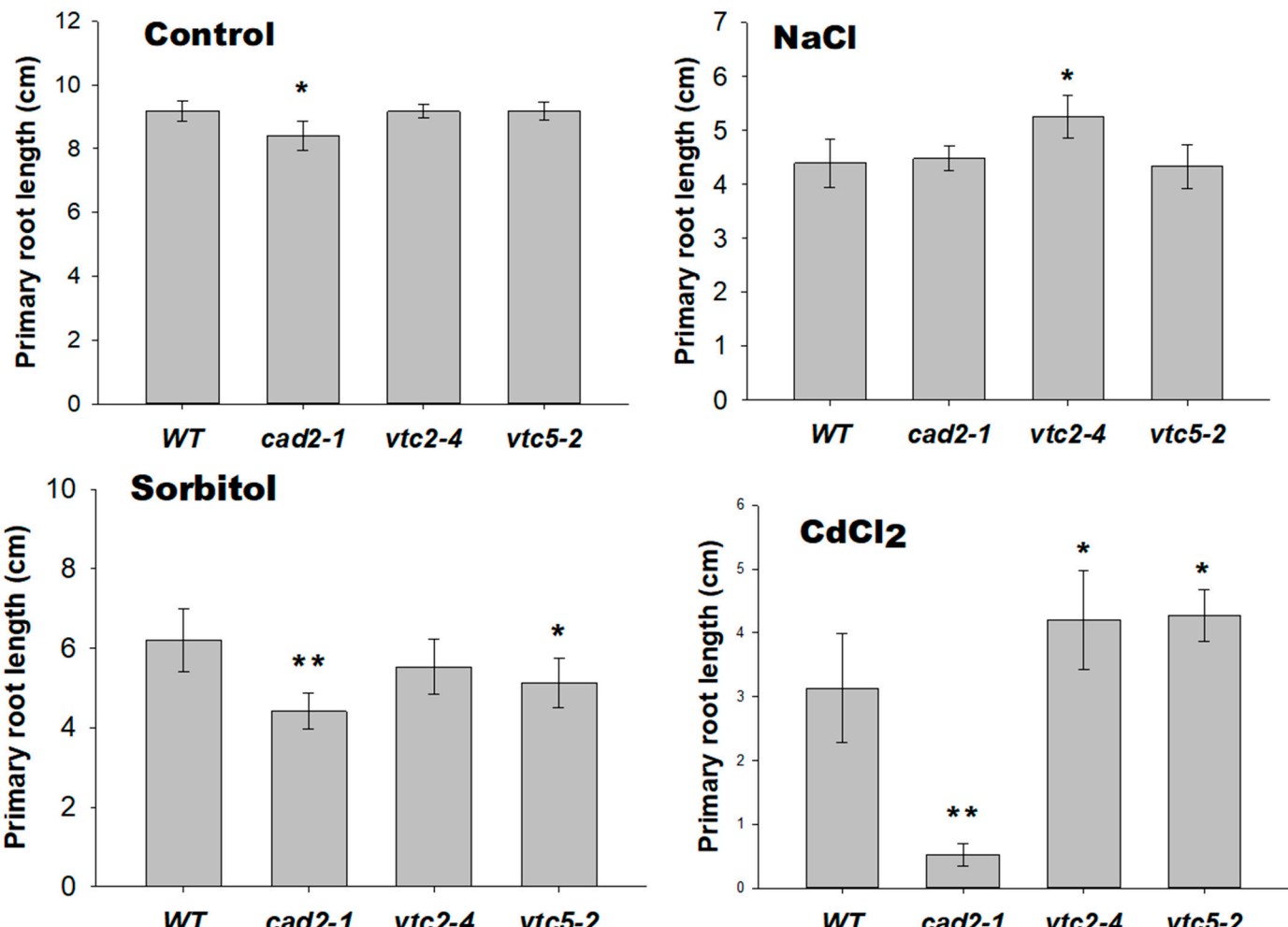

**Figure 7.** Primary root length of the *cad2-1*, *vtc2-4*, *vtc5-2* mutants and WT under normal conditions or different strong stress conditions. The *cad2-1*, *vtc2-4* and *vtc5-2* mutants as well as the WT seeds were exposed to different strong abiotic stress conditions: salt stress (100 mM NaCl), osmotic stress (225 mM sorbitol), oxidative stress (1 mM $H_2O_2$), and heavy metal stress (40 µM $CdCl_2$) for 18–21 days. Data are shown as the means ± SD (*n* = 5–10). Experiments were performed twice with similar results. Asterisks indicated the statistical significance of differences between WT and mutants by Student's *t*-test: * $p < 0.05$ and ** $p < 0.01$.

Under control conditions, the primary root lengths of the mutants were almost indistinguishable from the WT, while the *cad2-1* and *vtc2-4* mutants had lower numbers of LRs compared to the WT. Primary root elongation and lateral root production were significantly decreased in all stressed conditions compared to the unstressed conditions. Compared to the WT, the *cad2-1* mutants displayed no difference in primary root growth and lateral root production in medium with 100 mM NaCl. However, under these conditions, *vtc2-4* mutant showed an enhanced root system with longer primary roots and more lateral roots, which may help this mutant to enhance their salt stress tolerance. Thus, the *vtc2-4*

mutant was less sensitive to salt stress than the WT in primary root growth, lateral root production and leaf growth. On $\frac{1}{2}$ MS medium with 225 mM sorbitol, the lengths of *cad2-1* and *vtc5-2* primary roots were significantly decreased, whereas the length of *vtc2-4* primary roots was not different to the WT. Compared to the WT, the numbers of LRs of the *vtc2-4* and *vtc5-2* mutants were significantly increased, whereas the number of the *cad2-1* was not changed; when plants were grown on medium with 225 mM sorbitol. When grown on $\frac{1}{2}$ MS medium containing 40 µM CdCl$_2$, the primary roots of *cad2-1* was significantly shorter, while primary roots of *vtc2-4* and *vtc5-2* were longer than those of WT. The current study observed no significant difference in LRs production in *vtc2-5* and WT under CdCl$_2$ treatment. However, LRs number was markedly decreased in the *cad2-1* mutant, while it was significantly increased in the *vtc2-4* mutant in comparison to the WT. Primary root elongation of all genotypes was inhibited by supplementing 1 mM H$_2$O$_2$ (Supplementary Figure S2). These results suggested that GSH deficiency in plants altered the root architecture under unstressed conditions and heavy metal stress while AsA deficiency in plants altered the root architecture under strong salt, osmotic and heavy metal stresses. Taken together, the AsA and GSH-deficient mutants exposed to long-term strong abiotic stresses displayed pleiotropic effects on root architecture.

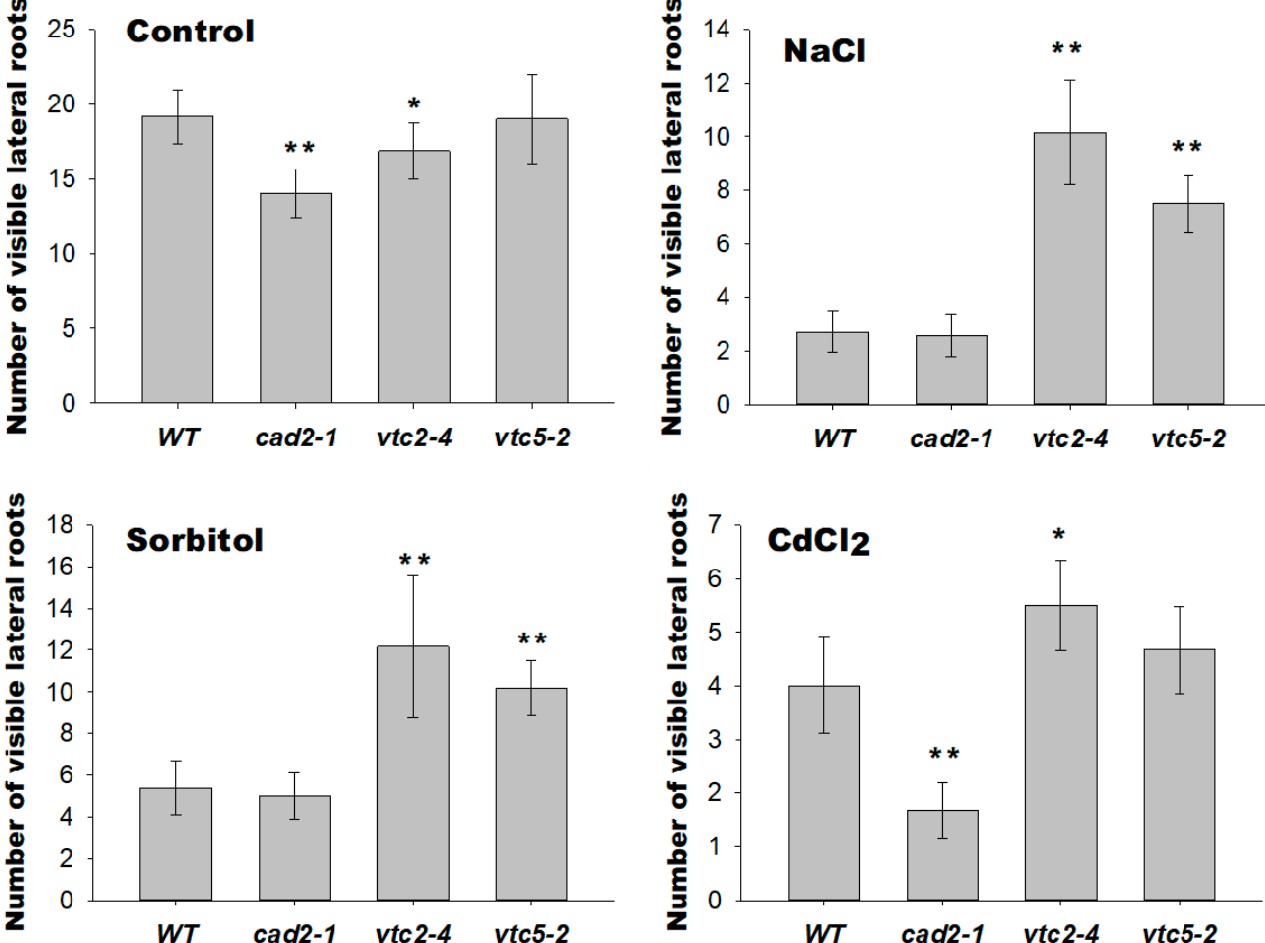

**Figure 8.** Lateral root number of the *cad2-1*, *vtc2-4*, *vtc5-2* mutants and WT under normal conditions or different strong stress conditions. The *cad2-1*, *vtc2-4* and *vtc5-2* mutants as well as the WT seeds were exposed to different strong abiotic stress conditions: salt stress (100 mM NaCl), osmotic stress (225 mM sorbitol), oxidative stress (1 mM H$_2$O$_2$), and heavy metal stress (40 µM CdCl$_2$) for 18–21 days. Data are shown as the means ± SD ($n$ = 5–10). Experiments were performed twice with similar results. Asterisks indicated the statistical significance of differences between WT and mutants by Student's *t*-test: * $p < 0.05$ and ** $p < 0.01$.

### 2.6. Plants with Lower AsA and GSH Levels Showed a Shorter Primary Root under Severe Abiotic Stresses

To further characterize and evaluate the responses of *cad2-1*, *vtc2-4* and *vtc5-2* after germination and seedling stages to different abiotic stresses under severe stress conditions, a transference system was applied. Arabidopsis seeds were sown and germinated on plates with $\frac{1}{2}$ MS media, then six-day-old seedlings of WT and mutants were transferred to the media supplemented with 150 mM NaCl, 375 mM sorbitol, 1.5 mM $H_2O_2$ or 100 μM $CdCl_2$, and primary root length was measured after five days. Under normal growth conditions, the primary root growth of the mutants was almost indistinguishable from the WT (Figure 9). When seedlings were transferred to severe stress conditions, primary root growth of *cad2-1*, *vtc2-4* and *vtc5-2* mutants was significantly more inhibited than the WT (Figure 9). These data suggest that AsA and GSH contents in plants are important for the primary root growth under severe stress conditions.

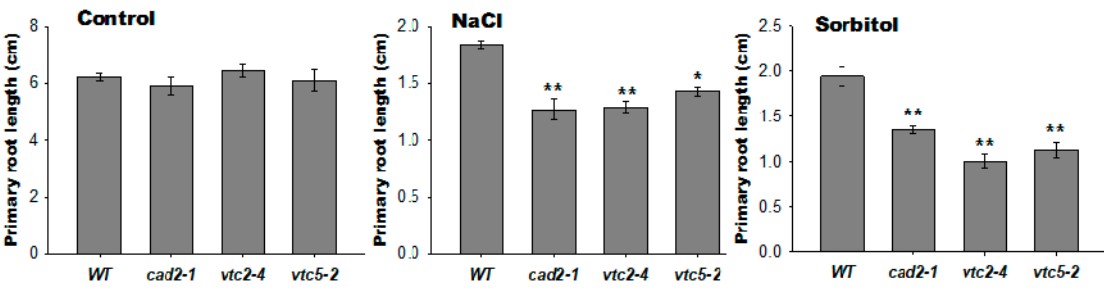

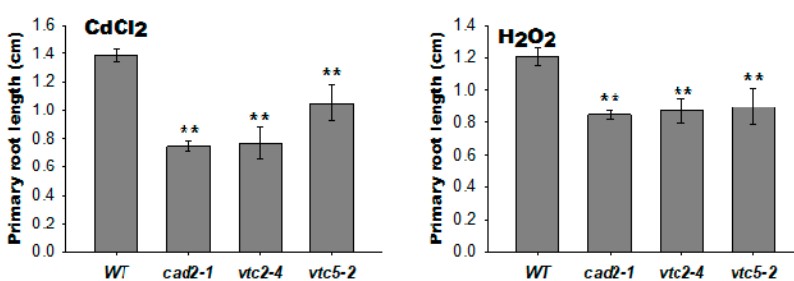

**Figure 9.** Primary root length of the *cad2-1*, *vtc2-4*, *vtc5-2* mutants and WT under normal conditions or different severe stress conditions. The *cad2-1*, *vtc2-4* and *vtc5-2* mutants as well as the WT seeds were grown on $\frac{1}{2}$ MS medium for 6 days. Six-day old seedlings were transferred to different severe abiotic stress conditions: salt stress (150 mM NaCl), osmotic stress (375 mM sorbitol), oxidative stress (1.5 mM $H_2O_2$), and heavy metal stress (100 μM $CdCl_2$) for 5 days and then photographed. Data are shown as the means ± SD (n = 5–7). Experiments were performed twice with similar results. Asterisks indicated the statistical significance of differences between WT and mutants by Student's *t*-test: * $p < 0.05$ and ** $p < 0.01$.

## 3. Discussion

This study demonstrates the importance of AsA and GSH during seed germination, root elongation in severe salt and osmotic stress conditions. However, the GSH contents did not affect the plant tolerance in strong salt stress conditions (100 mM NaCl) for long-term exposures. Deficiency of AsA enhanced salt stress tolerance in plants exposed to long-term strong salt stress as determined by leaf area, root elongation and lateral root development. GSH-deficient mutant plants increased their sensitivity to 40 μM $CdCl_2$ and 1.0 mM $H_2O_2$ for long-term exposures, suggesting the importance of GSH contents on detoxification of high $CdCl_2$ and $H_2O_2$. The AsA-deficient *vtc2-4* and *vtc5-2* mutants, however, decreased sensitivity toward $CdCl_2$ in term of leaf area. This study suggests that AsA and GSH may have various functions in abiotic stress tolerance and detoxification of ROS in plants, but their functions are dependent on stress-inducing agents and stress levels.

Antioxidant enzymes, such as catalases, superoxide dismutases and peroxidases, are the main ROS scavengers in plants [3]. Moreover, plants also possess antioxidant molecules, GSH and AsA, which effectively scavenge ROS directly and indirectly through enzymatic reactions [4]. In addition, AsA plays a role in Fe uptake through $Fe^{3+}$ reduction and is an important cofactor of enzymes involved in auxin degradation and synthesis of plant hormones (ethylene, abscisic acid, gibberellins), as well as anthocyanins and glucosinolates [17]. Moreover, GSH is also important in the synthesis of phytochelatins and detoxification of heavy metals, GSH interacts with hormones, and its redox state triggers signal transduction [21]. Here, this study characterized the GSH-deficient mutant (*cad2-1*) and AsA-deficient mutants (*vtc2-4*, *vtc5-2*) during germination, leaf growth, photosynthetic activity and root development under different abiotic stresses and stress levels. Interestingly, the GSH-deficient mutant and AsA-deficient mutants displayed altered sensitivity to salt, osmotic, oxidative stresses and heavy metal toxicity, which was depend on the stress levels. Therefore, this study suggested that GSH and AsA may play different roles in the tolerance to salinity, drought and Cd toxicity. These stresses can indirectly produce ROS, which leads to oxidative stress [33]. However, apart from the deleterious effects of oxidative damage, there is evidence that ROS are critical to and being continually produced during all phases of seed development, from desiccation to germination. Therefore, ROS may cover important roles in seed germination [34]. In fact, it has been shown that an optimal range of $H_2O_2$ is crucial for the dormancy release and a disturbance in ROS homeostasis decreases the seed germination [35,36]. Therefore, the balance between ROS production and scavenging should be strictly controlled during seed germination. Interestingly, *cad2-1*, *vtc2-4* and *vtc5-2* are not affected under no-stress conditions, whereas those mutants show a strongly reduction in germination rate under salinity and osmotic stress. Moreover, the most likely explanation for the differences between both *vtc* mutants is that *vtc2-4* mutant only contains 20–30%, while *vtc5-2* still contains 80% of the WT AsA level [5,19]. These results suggest that GSH and AsA synthesis mutants are still able to partially regulate seed ROS levels in the absence of external stress. Therefore, GSH and AsA may be important ROS scavengers implicated in dormancy release and seed germination under salt and osmotic stress.

In addition to ROS, another important player in seed germination is ABA. Germination begins with the release of dormancy, which is controlled by ABA. ROS accumulation acts as a positive signal for dormancy release by altering the synthesis and signaling of ABA. It has been demonstrated that $H_2O_2$ accumulation in germinating seeds is associated with ABA degradation, likely through the activation of ABA catabolic enzyme (ABA-8′-hydroxylase) or by the direct oxidation of ABA as well as antagonize ABA signaling [35,37,38]. In contrast, multiple studies in germinated seeds have shown that the direct inhibition of ABA blocks ROS production in seeds [39]. OH$^\bullet$ also promotes dormancy release by contributing to the cell wall loosening required for germination [40]. Dry seeds accumulate GSH and a very low amount of AsA and the AsA is synthetized de novo upon dormancy release and during germination [34]. Germinating seeds of AsA-deficient and GSH-deficient mutants very likely produce lower levels of AsA and GSH, respectively, compared to WT. Therefore, the disruption of GSH and AsA synthesis could reduce ABA-sensitivity during seed germination by the presence of high ROS levels, which may also alter the ABA homeostasis and signaling. However, the detailed molecular mechanism still remains unclear. Hence, further studies need to verify whether ROS and ABA levels are affected in GSH and AsA synthesis mutant seeds.

Regarding to seedling development, contradictory results related to GSH deficient mutants have been reported. While several studies showed that *cad2-1* mutant is significantly smaller than WT [22,41], recent works have reported that no distinct growth phenotype was observed between *cad2-1* and WT [20,25]. This study showed shorter primary roots and a decrease in lateral root numbers, but the leaf area showed no phenotypic difference between *cad2-1* mutant and WT seedlings. Likewise, previous experiments showed that *cad2-1* mutant is not affected by salt and osmotic stress [20,22]. However, in the present

study, the *cad2-1* exhibits strong sensitivity to osmotic stress. These pronounced differences emphasized the critical role of growth conditions. Indeed, GSH levels may vary significantly between experiments due to several non-controlled experimental differences [42]. Nevertheless, taking into account the importance of GSH in the root cell division [43] and auxin signaling [41,44], we speculate that the decrease of GSH causes an inhibition on root development by diminishing the cell proliferation and the auxin-dependent root growth. In contrast, under high ROS production provoked by osmotic stress, the inhibition of root elongation and enhancement of lateral roots might be related to the regulation of GSH concentration in the root apical meristem (RAM) and pericycle, respectively, which modulates the auxin signaling differently [44]. In the case of cadmium toxicity, GSH plays a key role as a chelator due to the high affinity of Cd for its thiol group, and also as a precursor for phytochelatins synthesis [31]. Therefore, a decreased capacity to chelate Cd ions due to a decrease in GSH levels and, by extent, low phytochelatins may contribute to the strong Cd sensitivity of *cad2-1* mutant.

Contrarily to *cad2-1*, *vtc2-4* and *vtc5-2* mutants were tolerant to 100 mM NaCl and 40 $\mu$M CdCl$_2$, but unaffected by sorbitol. AsA can potentially be oxidized and acts as a cofactor of 1-aminocyclopropane-1-carboxylic acid oxidase (ACO), the enzyme that catalyzes the last step of ethylene biosynthesis [45]. In addition, being a cofactor of dioxygenases, AsA could also be involved in the ABA biosynthesis by modulating the activity of the 9-cis-epoxycarotenoid dioxygenase (NCED3) responsible for the oxidative cleavage of neoxanthin to xanthoxin [17]. Recently, it was proposed that under salt stress, both ABA and ethylene production are induced, which could control ROS levels by regulating AsA biosynthesis through *VTC2* gene expression [46]. The *vtc2-4* and *vtc5-2* mutants could accumulate less ethylene and/or ABA, which could cause changes in plant response to abiotic stresses, leading to a decrease in growth inhibition at early stages. Intriguingly, the *vtc2-1* mutant accumulates more ABA and shows small size in non-stressed conditions [47]. However, it has been reported its small size is due to another mutation rather than VTC2 itself [5], suggesting that further analyses performed in different *vtc2* mutant alleles are required to determine how ethylene and ABA levels are affected in these mutant alleles.

Another hypothesis is based on the results observed in *vtc1-1* mutant, corresponding to a mutation on the gene *VTC1* encoded for GDP–d-Mannose pyrophosphorylase (VTC1), another AsA synthesis enzyme located upstream of VTC2 and VTC5 in the biosynthesis pathway [48]. Interestingly, this mutant accumulates higher amount of GSH than WT [49,50]. In addition, AsA deficiency was previously suggested to provide a primed state that decreases pathogen infection and abiotic stresses [51–53]. In addition to the elevated GSH levels, phytochelatin levels in *vtc1-1* were approximately twice in *vtc1-1* roots compared to WT plants upon Cd exposure [52]. However, mannose metabolism may be changed in this *vtc1-1* mutant, which might affect its physiological roles and responses to stress [54]. As one of the major defense mechanisms in Cd-exposed plants is chelation and sequestration by thiols [31], *vtc2-4* and *vtc5-2*, which are present also lower levels of AsA concentration, could also have higher GSH amount that increase the capacity to chelate Cd, contributing to a less Cd-sensitive phenotype.

Surprisingly, although *cad2-1*, *vtc2-4* and *vtc5-2* mutants are affected by oxidative stress, only *cad2-1* displays a stronger reduction in leaf area and lower Fv/Fm value. AsA and GSH are important players in the protection against ROS produced during photosynthesis [55]. Therefore, since AsA synthesis mutants may accumulate more GSH, these results also suggest that GSH is more active in the detoxification of ROS in chloroplasts, under strong oxidative stress, than AsA.

Photosynthesis is fundamental for plant growth. Photosynthetic activity of AsA-deficient mutants and GSH-deficient mutant was similar to WT at normal condition and under salt and osmotic stress. Under oxidative stress, Fv/Fm value was significantly reduced in *cad2-1* mutant while it was not changed in *vtc2-4* and *vtc5-2* mutants. Fv/Fm values of *cad2-1* mutant and *vtc2-4* mutant were decreased under Cd toxicity. Although AsA works as a reductant of violaxanthin de-epoxidase (VDE) during photo-oxidative

stress and a protective role in photoinhibition in heat-stressed leaves [13]. The AsA and GSH accumulation in mutants is possible to change their level in subcellular compartment such as chloroplasts, which might affect their photosynthetic activity [56].

Although the root growth of AsA-deficient mutants and GSH-deficient mutant responded differently to abiotic stress conditions at strong level (100 mM NaCl, 225 mM sorbitol, 1 mM $H_2O_2$ or 40 μM $CdCl_2$), all mutants showed a significant decrease of primary root growth response to severe abiotic stress conditions (150 mM NaCl, 375 mM sorbitol, 1.5 mM $H_2O_2$ or 100 μM $CdCl_2$). These results suggested that 20 to 80% AsA or 20–30% GSH was sufficient to maintain normal plant physiological activity under control or under osmotic and salt stress at strong level. However, the AsA and GSH contents or ratio of reduced form to oxidized form might change in AsA-deficient mutants and GSH-deficient mutant, compared to WT under severe stress conditions, which strongly affected the root growth. Interestingly, the enhanced root system observed in AsA-deficient mutants when they were exposed to strong salt stress (100 mM NaCl). Hence, these mutants may be beneficial and significant importance for studies on salt tolerance of economically important crops in the future.

This study showed that AsA and GSH could have different functions in detoxification of ROS in plants. This may include different intracellular compartmentation of both molecules and the ratio of reduced form to oxidized form under different abiotic stress conditions. Moreover, the AsA-GSH pathway is one of the main defense systems to protect the plants from multiple abiotic stresses [57]. Both AsA and GSH are strong antioxidants and the modulation of their redox state may work as a sensitive machinery to assess the stress levels and fine-tune molecular responses. Thus, the alteration in the synthesis of one of these molecules will strongly affect the stability of AsA-GSH pathway, leading to changes in the tolerance to abiotic stresses. Hence, the effects of altered GSH-AsA homeostasis in plant seed germination and seedling development need to be examined in further details, which will investigate the concentration and redox state of GSH and AsA in single or double mutants defective in AsA-biosynthesis gene and GSH-biosynthesis gene in response to the single and combinatorial stresses.

## 4. Conclusions

This study suggested AsA and GSH may have different functions in plant abiotic stress tolerance and ROS detoxification. Both AsA and GSH are important for seed germination under salt and osmotic stresses. Limitations of GSH and AsA synthesis reduce ABA-sensitivity during seed germination. Furthermore, both AsA and GSH are crucial factors for the primary root growth under severe stresses. GSH alone is a key player in heavy metal detoxification. Interestingly, deficiency of AsA favored larger leaf areas, enhancer root systems in plants exposed to long-term strong salt stress. Taken together, the findings shed new light on the functions of AsA in salt stress tolerance of plants, showing that deficiency of AsA might enhance plant salt stress tolerance. Therefore, further investigations of altered GSH-AsA homeostasis in plants need to be addressed in detail.

## 5. Materials and Methods

### 5.1. Plants Materials and Growth Conditions

*A. thaliana* ecotype Columbia (Col-0), *cad2-1* [24], *vtc2-4* (SALK_146824) and *vtc5-2* (SALK_135468) [15] were kindly provided by Stephane Mari (BPMP Montpellier, France). Seeds were surface sterilized by soaking in 70% ethanol for 1 min and 2.5% NaClO for 5 min, then washed four times with sterilized water. Sterilized seeds were sown on half-strength Murashige and Skoog (MS) medium containing 1.0% sucrose and 0.8% agar (adjusted to pH 5.8 with MES-KOH) ($\frac{1}{2}$ MS medium) [58]. Seeds were stratified for two days at 4 °C and then transferred to a growth chamber at 22 °C with light intensity of 120 μE m$^{-2}$ s$^{-1}$ and a 16 h light/8 h dark cycle [59]. Each experiment was repeated at least twice times.

### 5.2. Seed Germination Test

For germination assay, sterilized seeds of WT and *cad2-1* were sown on half strength MS medium containing 0.8% agar supplemented with or without NaCl (0, 50, 100, 150, 170, 200 mM), sorbitol (0, 200, 300, 400, 500, 600 mM) [60]. Sterilized seeds of WT, *vtc2-4* and *vtc5-1* were sown on the $\frac{1}{2}$ MS media supplemented with 150 mM NaCl or with 300 mM sorbitol. Sterilized seeds of WT, *cad2-1*, *vtc2-4* and *vtc5-1* were sown on $\frac{1}{2}$ MS media supplemented with different concentrations of ABA (0, 2.5, 5, 10 or 15 μM) [58]. The plates were kept at 4 °C for two days and then placed horizontally at growth chamber at 22 °C under a 16 h light and 8 h dark photoperiod. Germination rates were recorded at 2, 3, 4, 5, and 6 day, based on the radicle tip emergence.

### 5.3. Abiotic Stress Treatments

#### 5.3.1. Strong Long-Term Stress

Sterilized seeds were sown on plates in $\frac{1}{2}$ MS medium (control), or media containing 100 mM NaCl or 225 mM sorbitol, 1 mM $H_2O_2$, or 40 μM $CdCl_2$. Seedlings were grown for 18–20 days under these conditions [22,59]. The plates were scanned, shoot fresh weight was determined and the other analysis performed as described below.

#### 5.3.2. Severe Short-Term Stress

Seeds were sterilized and sown on MS $\frac{1}{2}$ media. Uniform 6-day-old seedlings were transferred into $\frac{1}{2}$ MS media or $\frac{1}{2}$ MS media supplemented with 150 mM NaCl or 375 mM sorbitol or 1.5 mM $H_2O_2$ or 100 μM $CdCl_2$ and grown for a further five days in vertically placed petri plate [22,61].

### 5.4. Phenotypic Analysis

#### 5.4.1. Leaf Area Measurement

For rosette leaf area measurements, the seeds were sown on $\frac{1}{2}$ MS agar plates containing the indicated stress treatment as described above and placed horizontally. Plants were grown at 22 °C with 16 h light/8 h dark photoperiod for 18–21 days. Total leaf area was measured and determined using the ImageJ software (https://imagej.nih.gov/ij/, accessed and downloaded on 19 May 2019) [22,59].

#### 5.4.2. Primary Root Length and Number of Lateral Root

For long-term exposure to strong abiotic stress conditions, the seeds were sown on $\frac{1}{2}$ MS agar plates containing the indicated stress treatment as described above and placed vertically for 18–22 days. Primary root length and number of lateral roots were analyzed using the ImageJ software (https://imagej.nih.gov/ij/, accessed and downloaded on 19 May 2019) from the images of the plants at day 14 [22].

For short-term exposure to severe abiotic stress conditions, the seeds were germinated and grown on $\frac{1}{2}$ MS for six days, and six-day-old seedlings were transferred to different severe abiotic stress conditions: salt stress (150 mM NaCl), osmotic stress (375 mM sorbitol), oxidative stress (1.5 mM $H_2O_2$), and heavy metal stress (100 μM $CdCl_2$) for 5 days and then photographed. Primary root length was analyzed using the ImageJ software (https://imagej.nih.gov/ij/, accessed and downloaded on 19 May 2019) [61].

#### 5.4.3. Photosynthesis Fluorescence Measurements

Maximum PSII quantum yield [$F_v/F_m = (F_m - F_o)/F_m$] of rosette leaves was determined using FluorCam FC 800-O (Photon Systems Instruments) and photosynthetic activity was measured at $F_v/F_m$. 18–21 days old plants were dark-adapted for 30 min before measurements and all fluorescence measurements were performed in vivo at room temperature. A saturating light of 1000 μmol photons m$^{-2}$ s$^{-1}$ was applied to the measure the maximum fluorescence. The data were analyzed using the Fluorcam 7 software (https://fluorcams.psi.cz/, accessed and downloaded on 08 March 2019) [62].

### 5.5. Plasmid, Yeast Strains and Yeast Growth

The open reading frames (ORF) of AtGSH1 and AtGSH2 were amplified from Arabidopsis cDNA using the primers 5′ATGGCGCTCTTGTCTCAAGC3′ and 5′TTAGTACAGC AGCTCTTCGAAC3′ and primers 5′ATGGGCAGTGGCTGCTCTTC3′ and TCAAATCA-GATATATGCTGTCC, respectively. The amplified products were A-tailed and cloned into pGEM-T Easy vector (Promega), and sub-cloned into the yeast expression vector pDR195 at the NotI sites, resulting in the pDR195-AtGSH1 and pDR195-AtGSH2 constructs [63].

Yeast cells were grown at 30 °C in yeast extract peptone dextrose YPD medium (2% glucose, 2% tryptone, and 1% yeast extract) or synthetic defined (SD) medium (2% glucose, 0.7% yeast nitrogen base with ammonium sulfate, pH 5.5). WT BY4741 (*MATa; his3, leu2, met15, ura3*), gsh1Δ (*MATa; his3, leu2, met15, ura3*, gsh1::kanMX4), gsh2 Δ (*MATa; his3, leu2, met15, ura3*, gsh12::kanMX4)* were kindly provided by Dr. Léon Dirick (BPMP Montpellier). Yeast cells were transformed with pDR195 or pDR195-AtGSH1 construct or pDR195-AtGSH2, using the LiAc ssDNA/PEG method [64]. Drop assay was performed as previously described with minor modifications [62,65]. Transformed yeast cells were cultured in liquid SD medium containing amino acids without uracil (URA) overnight at 30 °C. Yeast cells were harvested by centrifugation at 2500 rpm for 4 min and washed twice with sterile water. The cells were resuspended and adjusted to $OD_{600\ nm} = 1$, and 8 μl of 10-fold serial dilutions were spotted onto 2% (*w/v*) agar plates containing SD-URA medium alone or supplemented 100 μM $CdCl_2$, 1 M NaCl or 1 M sorbitol. The plates were incubated at 30 °C for 3–4 days before being photographed.

### 5.6. Statistical Analysis

The data were statistically analyzed using Excel version 2010 and GraphPad Prism 7 program (https://www.graphpad.com/scientific-software/prism/, accessed and downloaded on 14 July 2020). Statistically significant differences were performed by Student's *t*-test (* $p < 0.05$; ** $p < 0.01$) or by ANOVA using the Tukey's Honestly Significant Difference (HSD) test [66].

**Supplementary Materials:** The following are available online at https://www.mdpi.com/article/10.3390/agronomy11040764/s1, Figure S1: The effects of long-term exposures to different abiotic stress treatments on fresh weight of *cad2-1*, *vtc2-4* and *vtc5-2* mutants, Figure S2: The effects of long-term exposures to different abiotic stress treatments on root architecture of *cad2-1*, *vtc2-4* and *vtc5-2* mutants, Table S1: Seeds of *Arabidopsis thaliana cad2-1*, *vtc2-4* and *vtc5-2* are more resistant to ABA-mediated inhibition of seed germination.

**Author Contributions:** Conceptualization and Supervision, M.T.T.H.; formal analysis, methodology, software and validation, data curation, investigation, writing—original draft preparation M.T.T.H., M.T.A.D. and T.N.; writing—review and editing, M.T.T.H.; visualization, D.-P.T.; data curation, T.N.C.; project administration, and funding acquisition, P.N.D.Q. and T.P.T.D. All authors have read and agreed to the published version of the manuscript.

**Funding:** This research was funded by Vietnam National University-Ho Chi Minh City (VNU-HCM) under Grant Number NV2018-18-03 to Thi Phuong Thao Dang.

**Institutional Review Board Statement:** Not applicable.

**Informed Consent Statement:** Not applicable.

**Data Availability Statement:** Data sharing not applicable.

**Acknowledgments:** We would like to thank Stephane Mari (BPMP, Montpellier, France) for providing *Arabidopsis thaliana* Col-0 and line mutants including *vtc2-4* and *vtc5-2*, Léon Dirick (BPMP, Montpellier, France) for providing yeast mutant strains. We also thank Santiago Alejandro (Martin Luther University, Halle-Wittenberg, Germany), Tuan Minh Tran (Nanyang Technological University, Singapore), Thanh-Hao Nguyen (University of Science, VNU, Ho Chi Minh City, Vietnam) and Ngoc Hoang Bao Bui (University Health Network, Toronto, ON, Canada) for critical reading, editing, valuable suggestions and comments on improving the quality of the article.

**Conflicts of Interest:** The authors declare no conflict of interest.

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
