# Peer review of "Phenotypic Characterization of Arabidopsis Ascorbate and Glutathione Deficient Mutants under Abiotic Stresses"

_agronomy, doi:10.3390/agronomy11040764_

Round 1

Reviewer 1 Report

Despite considerable research on stress biology, the developmental functions of ascorbate and glutathione in plants are less well understood. This study evaluates developmental phenotypes of two Arabidopsis mutants deficient in ascorbate and one mutant deficicient in glutathione in non- and stressed conditions. There was a differential sensitivity of several vegetative growth parameters to salt, osmotic, H2O2 and cadmium stress. Perhaps the most interesting data were the relative insensitivity of germination of the ASC/GSH deficient mutants to abscisic acid (cf control). 

Major comments

  1. The introduction notes contradictions in function of Arabidopsis mutants deficient in ASC or GSH but the present study only includes mutants of a single allele of VTC2, VTC5 and GSH1. This is not a sufficient experimental design to strengthen the literature on the role of ASC or GSH deficiency on plant development or stress resistance.
  2. Following above, why were experiments not performed with or without addition of GSH or ASC to the media? Or precursors e.g. L-Gal or L-Cys? Or with BSO to inhibit GSH1? 
  3. It is unclear what the biological unit was for each experiment. Line 473-4 states each experiment was independently repeated three times but figure legends e.g. Fig 9 state n = 5-10. Most fig legends do not state n, nor do the methods. Furthermore there is no statistic data for several figures (e.g. Fig. 1-4) and yet the authors state significant differences. 
  4. The interpretation of results (Discussion) focussed almost entirely on antioxidant functions. ASC and GSH are not only involved in ROS detoxification and I find this a particular weakness. It is interesting that the ASC mutants are less sensitive to ABA than control considering tha that dry seed are almost completely devoid of ASC and require de novo synthesis upon imbibition. I do not believe this is adequately discussed (if the data are reliable).

Minor comments

  1. Line 400 tolerant should be sensitive or intolerant.
  2. Line 192 Abscicic - spelling error.

Author Response

Dear Reviewer,

We thank the reviewer for the positive comments, constructive suggestions and pointing out the weakness in our manuscript. The major points raised by the reviewer are explained and addressed as follow.

Major comments

  1. The introduction notes contradictions in function of Arabidopsis mutant deficient in ASC or GSH but the present study only includes mutants of a single allele of VTC2, VTC5 and GSH1. This is not a sufficient experimental design to strengthen the literature on the role of ASC or GSH deficiency on plant development or stress resistance.

Response:

We would like to thank the reviewer for pointing out that.

We are in full agreement with the reviewer on the lack of sufficiently experimental material to strengthen the literature on the role of ASC or GSH deficiency on plant development or stress resistance. However, there are contradictory results on the phenotypes of glutathione deficient mutants in response to abiotic stress conditions. Therefore, we agree that using the single allele mutant of GSH1 is a weak point in our research. As mentioned on the introduction, the functions of ascorbate on abiotic stress tolerance had used the EMS mutants such as vtc2-1 or vtc1-1 might contain the cryptic mutations which generate the pleiotropic effect. This study has used two new T-DNA insertion mutants (vtc5-2 and vtc2-4) with moderate and very low ascorbate contents in Arabidopsis to compare the effect of alteration of ascorbate levels on abiotic stress tolerance. To clarify this point, we provided more details in the introduction, page 2 as following ‘Therefore, the present study was designed to explore further the effects of altered endogenous AA on germination, shoot growth, root development and photosynthesis in response to abiotic stress conditions by phenotyping of Arabidopsis AsA-deficient mutants with moderate (vtc5-2) or very low (vtc2-4) AsA content, compared to WT. ‘

Moreover, one of our objectives is comparing the phenotypes of GSH-deficient mutants and AsA.-deficient mutants in response to different abiotic stress condition. Therefore we have further clarified this point in our objective in Introduction, page 3 as following ‘In addition, although AsA and GSH are both main important antioxidants for AsA-GSH cycle, the GSH- and AsA-related biosynthetic pathways in the response to abiotic stress conditions used to be investigated separately.’

The appropriate approach to further elucidate response mechanism of both AsA and GSH to abiotic stress requires to perform experiments on double mutants defected in both AsA and GSH biosynthesis. Therefore, we have spent over two years trying to generate a double mutant (cad2-1 and vtc2-4). However, we could not obtain any homozygous double mutants in our laboratory. According to Jozefczaf et al. (2015), they have generated the double mutant (cad2-1 and vtc1-1). However, mannose metabolism may be changed in this vtc1-1 mutant, which might affect its physiological roles and responses to stress. We also tried to contact this group to request this double mutant. Unfortunately, we did not receive any response. Due to several limitations from our side including financial support and time constrain, we were unable to spend more time for screening homozygous line or request this double mutant for current manuscript. We will try to find solution to get this double mutant and understand better the complex function of AsA and GSH as well as the link changes in AsA and GSH status in response to abiotic stress in the future. We addressed this limitation in Discussion section, page 15, line 464 as following ‘Hence, the effects of altered GSH-AsA homeostasis in plant seed germination and seedling development need to be examined in further details, which will focus on the concentration and redox state of GSH and AsA in single or double mutants defective in AsA-biosynthesis gene and GSH-biosynthesis gene in response to the single and combinatorial stresses.’

  1. Following above, why were experiments not performed with or without addition of GSH or ASC to the media? Or precursors e.g. L-Gal or L-Cys? Or with BSO to inhibit GSH1? 

Response:

Thank you for this suggestion. It is good idea to apply chemical or pharmacological approach and genetic approach at parallel moment.  We agree that it will be better if we can further perform our experiments with application of precursors or inhibitor compounds. We agree that exogenous application of GSH or AsA or their precursors are interesting. However, we needed to add new material to address other comments specifically about the genetics, which is the focus of this study. We have not been able to perform more experiments about exogenous application in the revised manuscript.

 As mentioned on introduction, some previous studies showed the application of exogenous AsA or GSH can improve tolerance against abiotic stresses by enhancing plant growth and rate of photosynthesis. However, how biosynthetic pathway of AsA or GSH is regulated in plants under stress and how far AsA or GSH accumulation in plants under stress is required further investigation. Therefore, our research mostly focused on the analysis of effect of abiotic stress on the genetic impairment of AsA or GSH production.

GSH deficiency resulting from the application of the GSH-synthesis inhibitor, buthionine sulfoximine (BSO) totally impairs root development in wild-type plants. Therefore, application of BSO will limit the analysis of abiotic stresses on root developments.

Combining above reasons, we hope the reviewer can understand and accept for us.

  1. It is unclear what the biological unit was for each experiment. Line 473-4 states each experiment was independently repeated three times but figure legends e.g. Fig 9 state n = 5-10. Most fig legends do not state n, nor do the methods. Furthermore there is no statistic data for several figures (e.g. Fig. 1-4) and yet the authors state significant differences. 

Response:

Due to the problem of word format in my computer, the figure legends (Fig. 2, 5, 6, 7) were changed to the text of the results in previous manuscript. We have modified their format. Our figure legends included all biological units for each figure. Each experiment was independently repeated at least twice times while the number of biological samples is different (n ranged from 5 to 100).

We have modified and provided statistical analysis for Figure 2 to Figure 4.  

  1. The interpretation of results (Discussion) focussed almost entirely on antioxidant functions. ASC and GSH are not only involved in ROS detoxification and I find this a particular weakness. It is interesting that the ASC mutants are less sensitive to ABA than control considering tha that dry seed are almost completely devoid of ASC and require de novo synthesis upon imbibition. I do not believe this is adequately discussed (if the data are reliable).

Response:

We are in agreement with the reviewer about these two points. First, we have added some sentences in the beginning of Discussion section, page 12 and 13, which explaining that AsA and GSH are also important in different plant metabolic processes, as follow “In addition, AsA plays a role in Fe uptake through Fe3+ reduction and is an important cofactor of enzymes involved in auxin degradation and synthesis of plant hormones (ethylene, abscisic acid, gibberellins), as well as anthocyanins and glucosinolates [17]. Moreover, GSH is also important in the synthesis of phytochelatins and detoxification of heavy metals, GSH interacts with hormones, and its redox state triggers signal transduction [21]. Moreover, we already mentioned that GSH could be involved in cell division, auxin signaling, heavy metal chelator and precursor of phytochelatin biosynthesis, as well as AsA could act as cofactor of 1-aminocyclopropane-1-carboxylic acid oxidase (ACO and 9-cis-epoxycarotenoid dioxygenase (NCED3) in th Discussion section at page 13. Second, we have added a short text to clarify that the function of AsA in seeds under ABA treatment would be after dormancy release, since dry seeds only accumulate very low levels of AsA. Therefore, AsA mutants would synthetize de novo low AsA after dormancy release compare to WT. We have add this part in Discussion section at page 13 as follow “Dry seeds accumulate GSH and a very low amount of AsA and the AsA is synthetized de novo upon dormancy release and during germination [34]. Germinating seeds of AsA-deficient and GSH-deficient mutants very likely produce lower levels of AsA and GSH, respectively, compared to WT.”

Minor comments

  1. Line 400 tolerant should be sensitive or intolerant.

Response: It is possible that our writing is not clear leading to your misunderstanding at this point. Therefore, we have added some other sentences to discusion to clarify this point as follow: “Interestingly, the enhanced root system observed in AsA-deficient mutants when they were exposed to strong salt stress (100 mM NaCl).”

  1. Line 192 Abscicic - spelling error.

Response: We have modified and changed to “Abscisis”

We thank you for your insightful comments.

Sincerely yours,

Minh Hoang

Reviewer 2 Report

The study aims to investigate the role of ascorbic acid (AsA) and glutathione (GSH) in seed germination, plant performance and abiotic stress tolerance by characterizing Arabidopsis GSH deficient cad2-1, and AsA deficient vtc2-4 and vtc5-2 mutants at seed germination and seedling stages. Though the approach is not innovative, the experiments are well designed and the results was performed in good manner.

To conclude, I think this manuscript can be published with average impact on the field. However, the article can be improved if authors can apply the same stress level for all genotypes or explain why they chose different concentrations; measure AsA and GSH concentration in parallel and interpreter more the different behaviors of the two mutants vtc2-4 and vtc5-2.

---------------------------------------------------

  1. What is the main question addressed by the research?

Authors have investigated on detailed analysis of the seed germination, shoot growth, root growth and photosynthetic activity of glutathione-deficient cad2-1, and ascorbate-deficient vtc2-4 and vtc5-2 mutants Arabidopsis plants under stress conditions. Their results demonstrated the functions of AsA and GSH in abiotic stress tolerance and detoxification of ROS in plants; however their roles are dependent on stress-inducing agents and stress levels.

  1. Is it relevant and interesting? Yes
  2. How original is the topic?

The functions of ascorbate on abiotic stress had studied using the EMS mutants such as vtc2-1 or vtc1-1 might contain the cryptic mutations which generate the pleiotropic effect. Therefore, this study has used two new T-DNA insertion mutants (vtc5-2 and vtc2-4) produce moderately and low ascorbate contents in Arabidopsis to compare the effect of ascorbate levels on abiotic stress conditions. Furthermore, there are several contradictory results on the phenotypes of glutathione deficient mutants in response to abiotic stress conditions. This research addressed and evaluated the functions of ascorbate and glutathione in different developmental stages and photosynthetic activity in response to abiotic stress conditions.

  1. What does it add to the subject area compared with other published material?

This research has focused on the response of two new T-DNA insertion mutants (vtc2-4 and vtc5-2) in different stress levels and stress-inducing agents. vtc2-4 and vtc5-2 mutants produced different response to NaCl, sorbitol and Cd toxicity in different stress levels. The low ascorbate concentration in ascorbate-deficient mutants might help plant growth better (improved root systems and bigger leaf areas) in 100 mM NaCl. These mutants may be of significant importance for studies on salt tolerance of economically important crops. These results will have impact on stress physiology.

  1. Is the paper well written? Yes
  2. Is the text clear and easy to read? Yes
  3. Are the conclusions consistent with the evidence and arguments presented? Yes
  4. Do they address the main question posed? Yes

Some minor comments:

  • Authors should use either “ascorbate-deficient” or “AsA- deficient” instead of “AsA reduction”
  • In figure legends: shouldn’t use the words “with similar results”
  • “sensitive to” instead of “sensitive with”
  • Explain more why they chose such different concentrations of stress-inducing agents
  • If possible, apply the same stress level for all genotypes; measure AsA and GSH concentration at the same growing conditions and interpreter more the different behaviors of the two mutants vtc2-4 and vtc5-2; use double mutants defected in both AsA and GSH biosynthesis.

Best regards,

Author Response

Dear Reviewer,

We thank the reviewer for the positive comments and constructive suggestions. The major points raised by the reviewer are explained and addressed as follow.

The study aims to investigate the role of ascorbic acid (AsA) and glutathione (GSH) in seed germination, plant performance and abiotic stress tolerance by characterizing Arabidopsis GSH deficient cad2-1, and AsA deficient vtc2-4 and vtc5-2 mutants at seed germination and seedling stages. Though the approach is not innovative, the experiments are well designed and the results were performed in good manner.

Response:

Thank you for your positive comments. We are so grateful for your kind words.

To conclude, I think this manuscript can be published with average impact on the field. However, the article can be improved if authors can apply the same stress level for all genotypes or explain why they chose different concentrations; measure AsA and GSH concentration in parallel and interpreter more the different behaviors of the two mutants vtc2-4 and vtc5-2.

Response:

Thank you for accepting our article to be published on Agronomy.

For following point “However, the article can be improved if authors can apply the same stress level for all genotypes or explain why they chose different concentrations”, we would like to explain and clarify as follow:

  1. Seed germinations of WT and cad2-1 were tested in different concentrations of NaCl or sorbitol to optimize the minimum inhibition dose for next experiment which compared the germination rates of WT, vtc2-4 and vtc5-2 mutants seeds in response to salt or osmotic stresses. Overall, the germinations of all genotypes were checked on control, 150 mM NaCl or 300 mM sorbitol.
  2. For long-term exposures to strong abiotic stresses, all genotypes were grown on 100 mM NaCl, 225 mM sorbitol, 40 µM CdCl2 or 1 mM H2O2 for 18-22 days, which was mentioned on subsection 4.3.1 of Materials and Methods section at page 16.
  3. For short-term exposures to severe abiotic stresses, 6 days old seedlings of all genotypes were transferred to 150 mM NaCl, 375 mM sorbitol, 100 µM CdCl2 or 1.5 mM H2O2 for 18-22 days, which was mentioned on subsection 4.3.2 of Materials and Methods section at page 16.

For your comments about “measure AsA and GSH concentration in parallel and interpreter more the different behaviors of the two mutants vtc2-4 and vtc5-2”.

            We agree that it is critical to measure AsA and GSH concentration in parallel. Due to several limitations from our side including infrastructure and financial support, we were unable to perform this experiment for the current manuscript. Therefore, we have discussed and propose further investigation about this point in Discussion section at page 15 as follow: “Hence, the effects of altered GSH-AsA homeostasis in plant seed germination and seedling development need to be examined in further details, which will investigate the concentration and redox state of GSH and AsA in single or double mutants defective in AsA-biosynthesis gene and GSH-biosynthesis gene in response to the single and combinatorial stresses.”

            For interpreter more the different behaviors of the two mutants vtc2-4 and vtc5-2, we have mentioned their responses into different stresses and explained their phenotypes on Discussion section at page 13 as follow: “Moreover, the most likely explanation  for the differences between both vtc mutants is that vtc2-4 mutant only contains 20-30%, while vtc5-2 still contains 80% of the WT AsA level [5,19].”

  1. What is the main question addressed by the research?

Authors have investigated on detailed analysis of the seed germination, shoot growth, root growth and photosynthetic activity of glutathione-deficient cad2-1, and ascorbate-deficient vtc2-4 and vtc5-2 mutants Arabidopsis plants under stress conditions. Their results demonstrated the functions of AsA and GSH in abiotic stress tolerance and detoxification of ROS in plants; however their roles are dependent on stress-inducing agents and stress levels.

Response: Thank you for understanding our work.

  1. Is it relevant and interesting? Yes

Response: Thank you for your assessment our work.

  1. How original is the topic?

The functions of ascorbate on abiotic stress had studied using the EMS mutants such as vtc2-1 or vtc1-1 might contain the cryptic mutations which generate the pleiotropic effect. Therefore, this study has used two new T-DNA insertion mutants (vtc5-2 and vtc2-4) produce moderately and low ascorbate contents in Arabidopsis to compare the effect of ascorbate levels on abiotic stress conditions. Furthermore, there are several contradictory results on the phenotypes of glutathione deficient mutants in response to abiotic stress conditions. This research addressed and evaluated the functions of ascorbate and glutathione in different developmental stages and photosynthetic activity in response to abiotic stress conditions.

Response:  Thank you for understanding our work.

  1. What does it add to the subject area compared with other published material?

This research has focused on the response of two new T-DNA insertion mutants (vtc2-4 and vtc5-2) in different stress levels and stress-inducing agents. vtc2-4 and vtc5-2 mutants produced different response to NaCl, sorbitol and Cd toxicity in different stress levels. The low ascorbate concentration in ascorbate-deficient mutants might help plant growth better (improved root systems and bigger leaf areas) in 100 mM NaCl. These mutants may be of significant importance for studies on salt tolerance of economically important crops. These results will have impact on stress physiology.

Response: Thank you very much for your positive comments.

  1. Is the paper well written? Yes

Response: Thank you for your assessment of our study.

  1. Is the text clear and easy to read? Yes

Response: Thank you for your assessment of our research.

  1. Are the conclusions consistent with the evidence and arguments presented? Yes

Response: Thank you for your evaluation of our work.

  1. Do they address the main question posed? Yes

            Response: Thank you for your assessment of our study.

Some minor comments:

  • Authors should use either “ascorbate-deficient” or “AsA- deficient” instead of “AsA reduction”

Response: We have changed to AsA-deficient mutants.

  • In figure legends: shouldn’t use the words “with similar results”

Response: Each experiment was performed at least twice independent replicates. Our data were presented at means ± SD of every single independent replicate. They were not shown as the mean ± SD of three/twice independent experiments. Therefore, we would like to keep “with similar results” to show that our data were producible and similar trend in independent experiments. We hope that you can understand and accept this point.

  • “sensitive to” instead of “sensitive with”

Response: We have used “sensitive to” for this manuscript.

  • Explain more why they chose such different concentrations of stress-inducing agents
  • If possible, apply the same stress level for all genotypes; measure AsA and GSH concentration at the same growing conditions and interpreter more the different behaviors of the two mutants vtc2-4 and vtc5-2; use double mutants defected in both AsA and GSH biosynthesis.

Response: The appropriate approach to further elucidate the response mechanism of both AsA and GSH to abiotic stress requires to perform experiments on double mutants defected in both AsA and GSH biosynthesis. Therefore, we have spent over two years trying to generate a double mutant (cad2-1 and vtc2-4). However, we could not obtain any homozygous double mutants in our laboratory. According to Jozefczaf et al. (2015), they have generated the double mutant (cad2-1 and vtc1-1). However, mannose metabolism may be changed in this vtc1-1 mutant, which might affect its physiological roles and responses to stress. We also tried to contact this group to request this double mutant. Unfortunately, we did not receive any response. Due to several limitations from our side including financial support and time constrain, we were unable to spend more time for screening homozygous line or request this double mutant for current manuscript. We will try to find solution to get this double mutant and understand better the complex function of AsA and GSH as well as the link changes in AsA and GSH status in response to abiotic stress in the future. We addressed this limitation in discussion, page 15 as following “Hence, the effects of altered GSH-AsA homeostasis in plant seed germination and seedling development need to be examined in further details, which will focus on the concentration and redox state of GSH and AsA in single or double mutants defective in AsA-biosynthesis gene and GSH-biosynthesis gene in response to the single and combinatorial stresses.”

We thank you for your insightful comments.

Sincerely yours,

Minh Hoang

Reviewer 3 Report

There are few suggestions from Reviewer in attached file. Hope an improved version of manuscript 

Author Response

Dear Reviewer,

We are very grateful to the reviewer for the insightful comments and valuable suggestions that helped to improve this manuscript. The points raised by the reviewer are explained and addressed as follow.

  • Comment A1. Reviewer suggested changing the title as follow: Stress phenotyping of ascorbate and glutathione deficient Arabidopsis mutants

Response:

Thank you very much for modifying and changing this title. We agree that it is good to have a title as short as possible. We also agree that your suggested title is short, precise and cover the main idea of this manuscript; however, it is broader than what we have done. Stress phenotyping includes phenotyping of abiotic, biotic and nutrition stresses. This research focused on the response of ascorbate and glutathione deficient Arabidopsis mutants into abiotic stresses.  To be more precise and encapsulate the main message of this manuscript, we would like to modify this title as follow: “Phenotypic analysis of Arabidopsis ascorbate and glutathione deficient mutants under abiotic environmental stresses”. We hope you can understand and accept for our new title.

  • Comment A2. Reviewer suggested delete word “reduced” in line 11.

Response: We have deleted this word following your suggestion

  • Comment A3: Reviewer suggested change “protect” in line 12 to “combat”.

Response: Thank you for your suggestion. “Combat” is nice word. However, we would like to keep “protect” for this situation. We hope you can understand and accept for our opinion.

  • Comment A4: Reviewer suggested change “photosynthetic activity” in line 15 to “photosynthesis”

Response: We have used “photosynthesis” as your suggestion

  • Comment A5: Reviewer suggested delete words “conditions, including salt, drought, heavy metal and oxidative stresses” in lines 15-16

Response: We have deleted those words as your suggestion

  • Comment A6: Reviewer suggested rewrite and quantify result outcome in lines 15-26

Response: Thank you for your valuable suggestion. We have modified this part as follow: “GSH deficiency in the cad2-1 plants affected plant growth, root development and photosynthesis in plants exposed to strong drought, oxidative and heavy metal stress conditions. Plants with lower GSH did not show an increased sensitivity to strong salt stress (100 mM NaCl). In contrast, the mutants with lower AsA enhanced salt stress tolerance in the long-term exposures since they showed larger leaf areas, longer primary root and more lateral root number. Limitations on AsA or GSH synthesis had no effect on decreases in photosynthesis in plants exposed to long-term strong salt or drought stresses, whereas they effected on photosynthesis of mutants exposed to CdCl2.”

  • Comment A7: Reviewer suggested change “Taken together, our data suggest” in line 26 to “The current study suggested”

Response: Thank you for your suggestion. Changing has been made to this point as follow “the current study suggested”

  • Comment A8: Reviewer suggested write complete word “germination” in line 26 – not in fractions

Response: We would like to apologize for this trouble of word format. The fraction was generated due to the changing word format. We could not fix this problem. We hope you can understand for us.

  • Comment A9: Reviewer suggested change “temperature extremes and heavy metals” in line 33 to “Heat, and heavy metals toxicity”

Response: We agree with your suggestion to change “heavy metal” to “heave metals toxicity”. We would like to keep “temperature extremes” due to “temperature extremes” can cover heat and cold stresses.

  • Comment A10: Reviewer suggested change “we performed” in line 100 to “Considering all these points the current study had main objectives,”

Response: We partially agree with your suggestion, and then we have modified the following phase “this study was designed to”

  • Comment A11: Reviewer suggested change “different” in line 101 to “The test”

Response: Thank you for your suggestion. However, we think it is better that we should delete “different” in this sentence. We hope you can understand for us.

  • Comment A12: Reviewer suggested change “stress conditions” in line 102 to “stresses” and quantify which abiotic stress studied in bracket)

Response: We agree with your suggestion and we modified as follow “abiotic stresses (salt, drought, oxidative and CdCl2 toxicity)

  • Comment A13: Reviewer suggested shift the paragraph in lines 105-116 to discussion section.

Response: We understand that the introduction was presented for the reader to understand the background and our goals. Moreover, we wanted to engage the reader's interest. Therefore, we presented a brief summing up at the end of the introduction.  Therefore, we would like to keep this paragraph at Introduction section to engage reader’s interest. We hope you can understand and accept our writing style.

  • Comment A14: Reviewer suggested replace the word ‘we’ in line 123 with ‘The current study/ The present study’.

Response: We agree with your suggestion. We have replaced the word “we”, “our data” by either “this study” or “the current study/the present study” or “these results”.

  • Comment A15: Reviewer suggested change “we examined whether the” in line 125 to “the”

Response: We agree with your suggestion. We have modified as follow “this experiment was performed to determine whether.

  • Comment A16: Reviewer suggested change “our” in line 130 to “The current study”

Response: We have replaced “our” by “these”.

  • Comment A17: Reviewer suggested change “raised a question about” in line 142 to “find out”

Response: We would like to keep “raise a question about” because the heterologous observation motivated us to ask if AtGSH1 serves similar functions in response to stress in plants.

  • Comment A18: Reviewer suggested change “we” in line 145 to “the present study”

Response: We have changed as your suggestion.

  • Comment A19: Reviewer suggested change “we” in line 153 to “the present study”

Response: We have changed as your suggestion.

  • Comment A20: Reviewer suggested delete the lines 166-168

Response: The lines 166-168 belonged to the legend of Figure 2. Due to the problem of word format in my computer, the figure legends (Fig. 2, 4, 5, 6, 7) were changed to the text of the results in previous manuscript. We have modified their format.

  • Comment A21: Reviewer suggested add the reference of the study in line 174

Response: This “previous experiment” means the germination rates of WT and cad2-1 in this manuscript. Seed germinations of WT and cad2-1 were tested in different concentrations of NaCl or sorbitol to optimize the minimum inhibition dose (150 mM NaCl or 300 mM sorbitol) for next experiment which compared the germination rates of WT, vtc2-4 and vtc5-2 mutants seeds in response to salt or drought stresses.

  • Comment A22: Reviewer suggested change “Abscicic” in line 192 to “Abscisic”.

Response: We have corrected this spelling.

  • Comment A23: Reviewer suggested change “We, therefore” in line 193 to “Therefor the present study”

Response: We have changed as your suggestion.

  • Comment A24: Reviewer suggested change “Our results” in line 199 to “The current study”

Response: We have changed as your suggestion

  • Comment A25: Reviewer suggested rewrite the results and quantify values derived from the current result outcome in lines 206-208

Response: As mentioned above in comment A20, due to the problem of word format in my computer, the figure legends (Fig. 2, 4, 5, 6, 7) were changed to the text of the results in previous manuscript. The line 206-208 belonged to legends of figure 4.

  • Comment A26: Reviewer suggested change “we” in line 229 to “The current study”

Response: We have changed as your suggestion

  • Comment A27: Reviewer suggested add the which heavy metal in bracket in line 231

Response: We have added (CdCl2) in the text

  • Comment A28: Reviewer suggested change “we” in line 305 to “The current study”

Response: We have changed as your suggestion

  • Comment A29: Reviewer suggested change “we” in line 344 to “The current study”

Response: We have changed as your suggestion

  • Comment A30: Reviewer suggested change “we” in line 346 to “The current study”

Response: We have changed as your suggestion

  • Comment A31: Reviewer suggested change “we” in line 349 to “The current study”

Response: We have changed as your suggestion

  • Comment A32: Reviewer suggested change “we” in line 382 to “The current study”

Response: We have changed as your suggestion

  • Comment A33: Reviewer suggested change “we” in line 386 to “The current study”

Response: We have changed as your suggestion

  • Comment A34: Reviewer suggested add the source of seeds of these mutant and accession number also the details of source accessions from where these mutant derived also how was these mutants developed (method Chemical or physical) and source institute where these were developed in line 467.

Response: We have modified this part as follow “thaliana ecotype Columbia (Col-0), cad2-1 [24], vtc2-4 (SALK_146824) and vtc5-2 (SALK_135468) [15] were kindly provided by Stephane Mari (BPMP Montpellier, France).

  • Comment A35: Reviewer suggested add the valid reference of protocol 4.1

Response: We have added the reference number 58 and 59 for this protocol

  • Comment A36: Reviewer suggested add the valid reference of protocol 4.2

Response: We have added the reference number 60 for this protocol

  • Comment A37: Reviewer suggested quantify what were major abiotic stresses under this study

Response: We have added “abiotic” to clarify.

  • Comment A38: Reviewer suggested add the valid reference of protocol 4.3.1

Response: We have added the reference number 22 and 59 for this protocol

  • Comment A39: Reviewer suggested add the valid reference of protocol 4.3.2

Response: We have added the reference number 22 and 61 for this protocol

  • Comment A40: Reviewer suggested add the valid reference of protocol 4.4.1

Response: We have added the reference number 22, 51 for this protocol

  • Comment A41: Reviewer suggested add the the version and reference of ImageJ software

Response: We have added the link of this software as follow “(https://imagej.nih.gov/ij/)”

  • Comment A42 & A43: Reviewer suggested add the valid reference of protocol 4.4.2

Response: We have added the reference number 22 and 61 for this protocol

  • Comment A44: Reviewer suggested add the reference of the formula [Fv/Fm = (Fm – Fo)/Fm] in line 514

Response: We have added the reference number 62 for this protocol

  • Comment A45: Reviewer suggested add the reference of FluorCam FC 800-O (Photon Systems Instruments) in line 515

Response: We have added the reference link (https://fluorcams.psi.cz/) for this instrument.

  • Comment A46: Reviewer suggested add the valid reference of protocol 4.4.3

Response: We have added the reference number 62 for this protocol

  • Comment A47: Reviewer suggested add the reference of Fluorcam 7 software in line 519

Response: We have added the reference link (https://fluorcams.psi.cz/) for this software

  • Comment A48: Reviewer suggested add the valid reference of primer sequences in 4.5

Response: Those primer sequences were designed by us. Therefore, we could not provide the reference. 

  • Comment A49: Reviewer suggested add the valid reference of the study in line 527

Response: We have added the reference number 63 for this protocol

  • Commented A50: Reviewer suggested add the valid reference of the protocol in lines 528 – 532

Response: We have added the reference number 62 and 65 for this protocol

  • Commented A51: Reviewer suggested add the valid reference of the protocol in lines 534 – 540

Response: We have added the reference number 62 and 65 for this protocol

  • Commented A52: Reviewer suggested change “or GraphPad Prism 7 program” in line 542 to “and statistical analytical software”

Response: We have modified this part as follow “The data were statistically analyzed using Excel version 2010 or GraphPad Prism 7 program (https://www.graphpad.com/scientific-software/prism/). Statistically significant differences were performed by Student’s t-test (*P < 0.05; **P < 0.01) or by ANOVA using the Tukey’s Honestly Significant Difference (HSD) test [66]

  • Commented A53: Reviewer suggested add the reference of GraphPad Prism 7 software

Response: We have added the reference link (https://www.graphpad.com/scientific-software/prism/)

  • Commented A54: Reviewer suggested add the t-test were done for which traits and analysis in line 543.

Response: Statistically significant differences were performed by Student’s t-test (*P < 0.05; **P < 0.01) or by ANOVA using the Tukey’s Honestly Significant Difference (HSD) test [66]

We thank you for your insightful comments.

Sincerely yours,

Minh Hoang

Round 2

Reviewer 3 Report

Appreciate the author and team for revised manuscript but same time embarrassed also that many points still same. Reviewer add suggestions in attached file. Hope an improved version. 

Author Response

Dear Reviewer,

We are very grateful to the reviewer for the insightful comments and valuable suggestions that helped to improve our manuscript. The points raised by the reviewer are explained and addressed as follow.

Comment A1. Reviewer again suggested changing the title as follow:

 Stress phenotyping of ascorbate and glutathione deficient Arabidopsis mutants.

(Phenotypic analysis of Arabidopsis ascorbate and glutathione deficient mutants under abiotic environmental stresses)

Term phenotyping more appropriate in place of Phenotypic analysis

Here author and team studied about ascorbate and glutathione deficient mutants of Arabidopsis

There is no need to write abiotic environmental in title all environmental stress are abiotic so no need to repeat two term similar meaning

Whereas stress phenotyping have more sound inference than abiotic environmental stress

Also Researcher working on ascorbate and glutathione, already know this this comes under abiotic stress so no need to use term abiotic stress

Response:

Thank you for your suggestion and detailed explanation. We appreciate your effort and time in evaluating our manuscript and help us to improve this title.  

We agree that term “abiotic environmental” is redundant. Therefore, we will eliminate “environmental” to this term. However, environmental stresses are including abiotic, biotic stresses. This manuscript focused only the effect of abiotic stresses on the phenotypes of ascorbate and glutathione deficient Arabidopsis mutants. Therefore, we cannot eliminate “abiotic” on the title. Functions of ascorbate and glutathione are not only abiotic stress tolerance but they involves in biotic stress also. We mentioned this point in reference number 26 at Introduction section, page 3 and reference 53 at Discussion section, page 14.

As we mentioned at the letter response to reviewer at first round, the term “stress phenotyping” is broader than what we have done in our manuscript. Stress phenotyping includes phenotyping of abiotic, biotic and nutrient stresses.

The instruction for authors of Agronomy Journal has introduced “The title should be concise, specific and relevant”. The title should be informative, accurately describes the contents of manuscript, and makes people want to read further. Our manuscript is not only phenotyping but also analysis and characterization of mutants under abiotic stresses. Therefore, the term “stress phenotyping” does not give enough information about what makes our manuscript interesting. If we give details of our research design as the term “phenotypic analysis” or “phenotypic characterization”, it will be more precise and it will likely attract more readers to our manuscript.

After we considered all of above reasons and features, we would like to modify our manuscript as follow:

“Phenotypic analysis of Arabidopsis ascorbate and glutathione deficient mutants under abiotic stresses”

OrPhenotypic characterization of Arabidopsis ascorbate and glutathione deficient mutants under abiotic stresses”

              We hope you can accept our manuscript’s title and understand our points.

Comment A2. Reviewer suggested change word “show” in line 21 to “showed”.

Response:

The sentence in line 21 is “Plants with lower GSH did not show an increased sensitivity to strong salt stress (100 mM NaCl)”. This is a simple past negative sentence. Therefore, we cannot change word “show” to “showed” as your suggestion.

Comment A3: Reviewer suggested revise whole manuscript and write complete word together in either of line, such as “oxidative” in lines 45&46.

Response:

We would like to apologize for this trouble of word format. The fraction was generated due to the changing word format. We could not fix this problem. We hope you can understand us.

Comment A4: Reviewer suggested shift the paragraph in lines 121-133 to discussion section. no need to write here in introduction  (In introduction no need to discuss any result outcome)

Response:

We agree with your suggestion. We have shifted the paragraph in line 121-133 to discussion section. To fulfill the requirements of Agronomy journal for Introduction section, we have added a sentence at the end of the introduction to highlight the main conclusions as follow: “The findings demonstrate that AsA and GSH play various roles in plant abiotic stress tolerance, but their functions are dependent on stress-inducing agents and stress levels.”

Comment A5: Reviewer suggested add the reference of the “previous experiments” in line 194.

Response:

Unfortunately, this point was not clear in our original manuscript. We would like to apologize for this misunderstanding. In this context, “previous experiment” means the germination rates of WT and cad2-1 in this current manuscript. Seed germinations of WT and cad2-1 were tested in different concentrations of NaCl or sorbitol to optimize the minimum inhibition dose (150 mM NaCl or 300 mM sorbitol) for next experiment which compared the germination rates of WT, vtc2-4 and vtc5-2 mutant seeds in response to salt or drought stresses. We cannot add any reference for this “previous experiments” in line 194. To clarify this point, we have modified and changed into “above experiments”.

Comment A6: Reviewer suggested change word “or” in line 588 to “and”.

Response:

Thank you for your suggestion. We have changed “or” into “and”.

We thank you for your insightful comments.

We would be glad to respond to any further questions and comments that you may have.

Sincerely yours,

Minh Hoang
